# Individual cristae within the same mitochondrion display different membrane potentials and are functionally independent

Dane M Wolf[1,2,†], Mayuko Segawa[1,†], Arun Kumar Kondadi[3,‡], Ruchika Anand[3,‡], Sean T Bailey[4,5,6] (iD), Andreas S Reichert[3], Alexander M van der Bliek[7,8] (iD), David B Shackelford[4,5], Marc Liesa[1,7,*,§] (iD) & Orian S Shirihai[1,2,**,§] (iD)

## Abstract

The mitochondrial membrane potential ($\Delta\Psi_m$) is the main driver of oxidative phosphorylation (OXPHOS). The inner mitochondrial membrane (IMM), consisting of cristae and inner boundary membranes (IBM), is considered to carry a uniform $\Delta\Psi_m$. However, sequestration of OXPHOS components in cristae membranes necessitates a re-examination of the equipotential representation of the IMM. We developed an approach to monitor $\Delta\Psi_m$ at the resolution of individual cristae. We found that the IMM was divided into segments with distinct $\Delta\Psi_m$, corresponding to cristae and IBM. $\Delta\Psi_m$ was higher at cristae compared to IBM. Treatment with oligomycin increased, whereas FCCP decreased, $\Delta\Psi_m$ heterogeneity along the IMM. Impairment of cristae structure through deletion of MICOS-complex components or Opa1 diminished this intramitochondrial heterogeneity of $\Delta\Psi_m$. Lastly, we determined that different cristae within the individual mitochondrion can have disparate membrane potentials and that interventions causing acute depolarization may affect some cristae while sparing others. Altogether, our data support a new model in which cristae within the same mitochondrion behave as independent bioenergetic units, preventing the failure of specific cristae from spreading dysfunction to the rest.

**Keywords** crista junction; cristae; membrane potential; MICOS complex; Opa1
**Subject Category** Membranes & Trafficking
**The EMBO Journal (2019) 38: e101056**
See also: **M Schlame** (November 2019)

## Introduction

Mitochondria utilize nutrients and molecular oxygen to generate a membrane potential ($\Delta\Psi_m$) across the inner mitochondrial membrane (IMM). The energy available for ATP synthesis is directly derived from $\Delta\Psi_m$ (Mitchell, 1961; Mitchell & Moyle, 1969); therefore, depolarization directly translates to decreased energy availability for ATP synthesis.

Classical studies suggested that the $\Delta\Psi_m$ was homogeneous along the IMM. Data supporting the equipotential model are (i) mitochondria labeled with $\Delta\Psi_m$-dependent dyes show a homogeneous signal along a single mitochondrion visualized under a low resolution microscope, indicating that the $\Delta\Psi_m$ is likewise homogeneous all along the organelle (Amchenkova et al, 1988; Skulachev, 2001); and (ii) an elongated mitochondrion stained with a $\Delta\Psi_m$-dependent dye appears to instantaneously lose its $\Delta\Psi_m$ following laser-induced damage to a small ($\leq 0.5\ \mu m^2$) region, suggesting that a mitochondrial filament is analogous to a power cable, where, if one part is compromised, the voltage will simultaneously collapse across its entire length (Amchenkova et al, 1988; Skulachev, 2001; Glancy et al, 2015). These conclusions were drawn after imaging mitochondria with $\Delta\Psi_m$-dependent dyes performed with light microscopes lacking sufficient spatial resolution to visualize the ultrastructure of the IMM. Furthermore, previous studies lacked the temporal resolution to determine whether laser-induced depolarization leads to an instantaneous collapse of $\Delta\Psi_m$ across the whole organelle.

The IMM consists of subcompartments called cristae and inner boundary membrane (IBM) (Palade, 1953). Cristae are invaginations

1 Department of Medicine (Endocrinology), Department of Molecular and Medical Pharmacology, David Geffen School of Medicine, University of California, Los Angeles, CA, USA
2 Graduate Program in Nutrition and Metabolism, Graduate Medical Sciences, Boston University School of Medicine, Boston, MA, USA
3 Institute of Biochemistry and Molecular Biology I, Medical Faculty, Heinrich Heine University Düsseldorf, Düsseldorf, Germany
4 Department of Pulmonary and Critical Care Medicine, David Geffen School of Medicine, University of California, Los Angeles, CA, USA
5 Jonsson Comprehensive Cancer Center, David Geffen School of Medicine, University of California, Los Angeles, CA, USA
6 Lineberger Comprehensive Cancer Center, University of North Carolina at Chapel Hill, Chapel Hill, NC, USA
7 Molecular Biology Institute at UCLA, Los Angeles, CA, USA
8 Department of Biological Chemistry, David Geffen School of Medicine at UCLA, Los Angeles, CA, USA
*Corresponding author. Tel: +1-310-206-7319; E-mail: mliesa@mednet.ucla.edu
**Corresponding author. Tel: +1-617-230-8570; E-mail: OShirihai@mednet.ucla.edu
†These authors contributed equally to this work as first authors
‡These authors contributed equally to this work as second authors
§These authors contributed equally to this work as corresponding authors

protruding into the mitochondrial matrix, whereas the IBM runs parallel to the outer mitochondrial membrane (OMM). Cristae and IBM are connected via narrow tubular or slit-like structures, known as crista junctions (CJs). In recent years, studies show that components of the electron transport chain (ETC) are confined to the lateral surfaces of the cristae rather than equally distributed along the IMM (Vogel *et al*, 2006; Wilkens *et al*, 2013). Moreover, dimers of $F_1F_0$ ATP Synthase assemble in rows along the edges of the cristae (Dudkina *et al*, 2005; Strauss *et al*, 2008; Davies *et al*, 2011). The CJs can be kept in a closed state by oligomers of the inner-membrane dynamin-like GTPase, OPA1 (Frezza *et al*, 2006; Pham *et al*, 2016), as well as components of the mitochondrial contact site and cristae organizing system (MICOS complex) (John *et al*, 2005; Rabl *et al*, 2009; Barrera *et al*, 2016; Glytsou *et al*, 2016).

These findings provide a conceptual framework, where protons pumped by the ETC across the cristae membrane appear first in the cristae lumen (Busch *et al*, 2013; Pham *et al*, 2016). However, unless the ΔΨ of the crista membrane is kept more negative compared to its neighboring IBM, protons would not remain in the cristae. This consideration implies that differences in $\Delta\Psi_m$ would exist between the cristae membrane and the IBM.

To establish whether the distribution and structural properties of OXPHOS complexes are functionally significant, it would be critical to directly visualize and quantify the $\Delta\Psi_m$ in relation to the IMM in living cells. If the $\Delta\Psi_m$ is uniform from one end of a mitochondrion to another, the $\Delta\Psi_m$ would be equal at any point along the IMM, supporting the equipotential model. If, however, the $\Delta\Psi_m$ stems from cristae functioning as independent and heterogeneous compartments, the $\Delta\Psi_m$ would vary substantially along the IMM—between cristae and IBM, as well as between different cristae. Testing such hypotheses, nonetheless, has remained virtually unfeasible, because the only way to resolve the IMM has been with the electron microscope, which requires freezing or fixation of mitochondria and therefore precludes any direct measurement of structure and $\Delta\Psi_m$.

To overcome this limitation, we developed a novel approach for imaging the IMM at high spatiotemporal resolution in living cells, using the LSM880 with Airyscan as well as STED microscopy. Staining active mitochondria with various dyes, we verified that we can resolve cristae from IBM. We then used various $\Delta\Psi_m$-dependent dyes to explore how the intricate architecture of the IMM relates to the most basic mitochondrial function—the $\Delta\Psi_m$ generated by the electrochemical gradient of protons.

## Results

### Development of an Airyscan-based approach to resolve cristae and IBM in living cells

Previous studies show that components of OXPHOS are unevenly distributed between the cristae and IBM (Vogel *et al*, 2006; Wilkens *et al*, 2013), suggesting the possibility of $\Delta\Psi_m$ heterogeneity along the IMM within a single mitochondrion. To develop an approach for the imaging of $\Delta\Psi_m$ associated with cristae and IBM, we first sought to determine whether we could resolve the compartmentalization of the IMM in living cells. To address this question, we incubated various cell types with 10-*N*-nonyl acridine orange (NAO), a fluorescent probe that preferentially binds cardiolipin but also shows some affinity for other phospholipids found in mitochondria, such as phosphatidylethanolamine (PE) and phosphatidylinositol (PI) (Leung *et al*, 2014). Imaging mitochondria from living HeLa, L6, and H1975 cells with the LSM880 equipped with Airyscan technology, we resolved intramitochondrial structures, typically perpendicular to the long axis of the mitochondrion, resembling cristae, as observed in electron micrographs (Fig 1A). Accordingly, the high resolution of the Airyscan-based microscopy allowed separation of cristae structures, IBM, as well as dimmer regions, appearing to be matrix (Fig 1B). To verify that they were matrix, we used matrix-targeted DsRed to label the matrix in H1975 cells and stained their IMM with NAO (Fig 1C). Airyscan imaging confirmed that the dimmer regions of NAO fluorescence within the mitochondria showed the strongest matrix-DsRed signal and vice versa (arrowheads). To confirm that we correctly identified the cristae structures in cells stained with NAO, we tested whether the pattern of NAO labeling was changed in cells with disrupted cristae structure. As a model, we used HeLa cells with Crispr/Cas9-mediated KO of Mic13 (Fig EV1A), which destabilizes CJs and disrupts cristae structure (Fig 1D) (Anand *et al*, 2016; Guarani *et al*, 2015). Compared to control HeLa (Fig 1A and B), Mic13-KO mitochondria showed a substantial decrease in the number of perpendicular structures, supporting their identification as cristae. As a second model of cristae perturbation, we examined H1975 cells with stable KD of PTPMT1 through lentiviral transduction encoding shRNA (Fig EV1B). PTPMT1 is a mitochondrial phosphatase, essential for biosynthesis of phosphatidylglycerol, a precursor of cardiolipin. Deletion of PTPMT1 has been shown to result in severe derangement of the IMM (Zhang *et al*, 2011). Our Airyscan imaging of

---

**Figure 1. High-resolution fluorescence imaging using Airyscan resolves the inner mitochondrial membrane (IMM) structure in live cells.**

High-resolution imaging of mitochondria in live cells using the Airyscan module of Zeiss LSM880 confocal microscope.

A   Images of IMM in living HeLa, L6, and H1975 cells, stained with 10-*N*-nonyl acridine orange (NAO). NAO preferentially binds phospholipids in the IMM, such as cardiolipin. Arrowheads indicate cristae in the IMM. Scale bar = 500 nm. $N \geq 3$ independent experiments for each cell type.

B   Mitochondrion cropped from HeLa cell shown in (A) (dashed line) and zoomed in to show cristae, inner boundary membrane (IBM), and matrix (arrows). Scale bar = 100 nm.

C   H1975 cells transduced with matrix-targeted DsRed and stained with NAO. Matrix-targeted DsRed differentiates matrix from cristae stained with NAO. Arrowheads point to cristae. Scale bar = 500 nm. $N = 1$ independent experiment.

D   The structure of the IMM in Mic13-KO cells (HeLa), stained with NAO. The number of cristae is decreased in Mic13-KO compared to control cells shown in panel (A), labeled with arrowheads. Scale bar = 500 nm. $N = 3$ independent experiments.

E, F   Live-cell imaging of the IMM in control and PTPMT1 KD H1975 cells, a model of cardiolipin deficiency. (E) A gallery of mitochondria from various control H1975 cells expressing scrambled shRNA and stained with NAO, showing cristae (arrowheads). Scale bars = 500 nm. $N = 3$ independent experiments. (F) A gallery of mitochondria from various H1975 cells expressing PTPMT1 shRNA and stained with NAO. Note the derangement of the ultrastructure (arrowheads). Scale bars = 500 nm. $N = 3$ independent experiments.

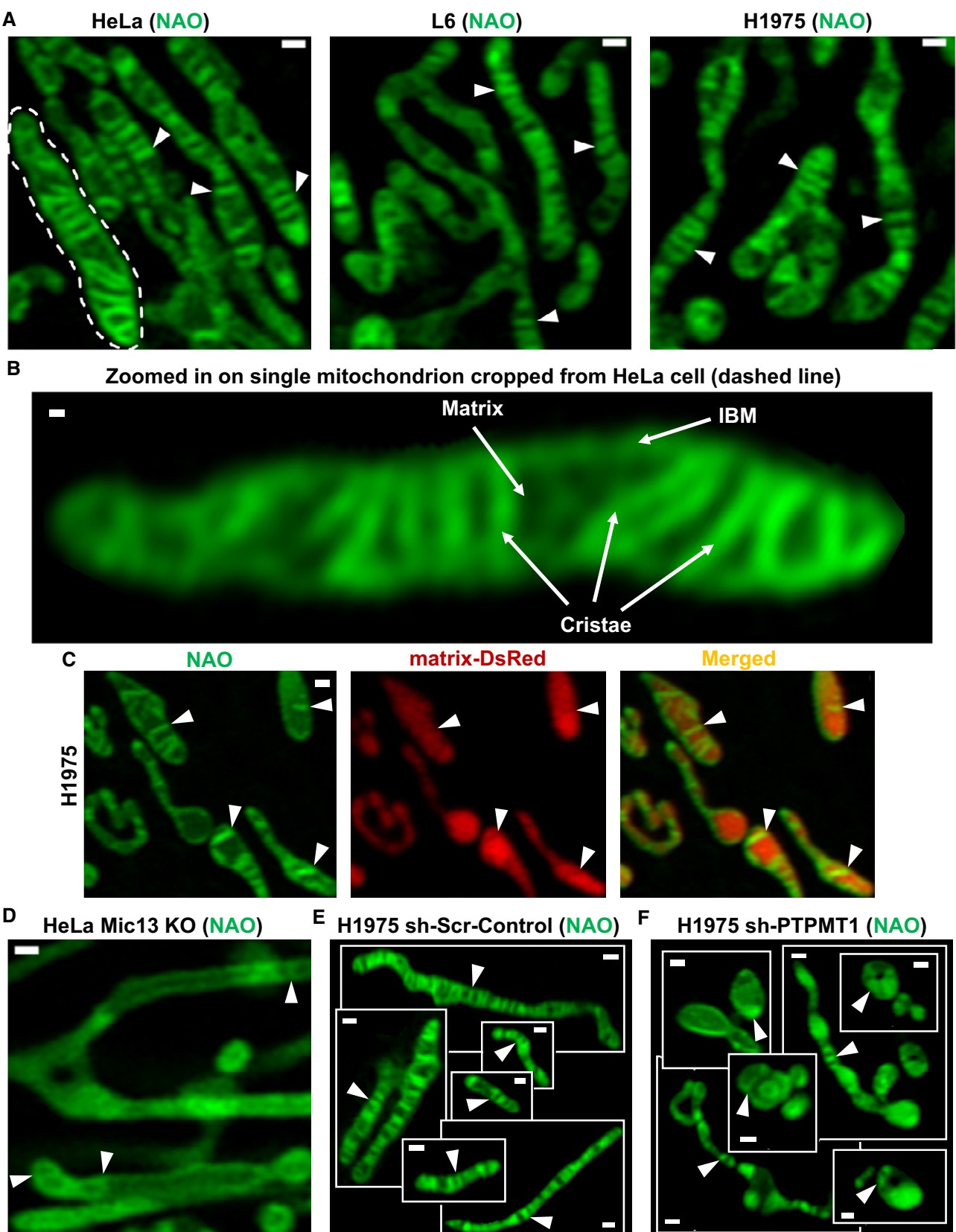

**Figure 1.**

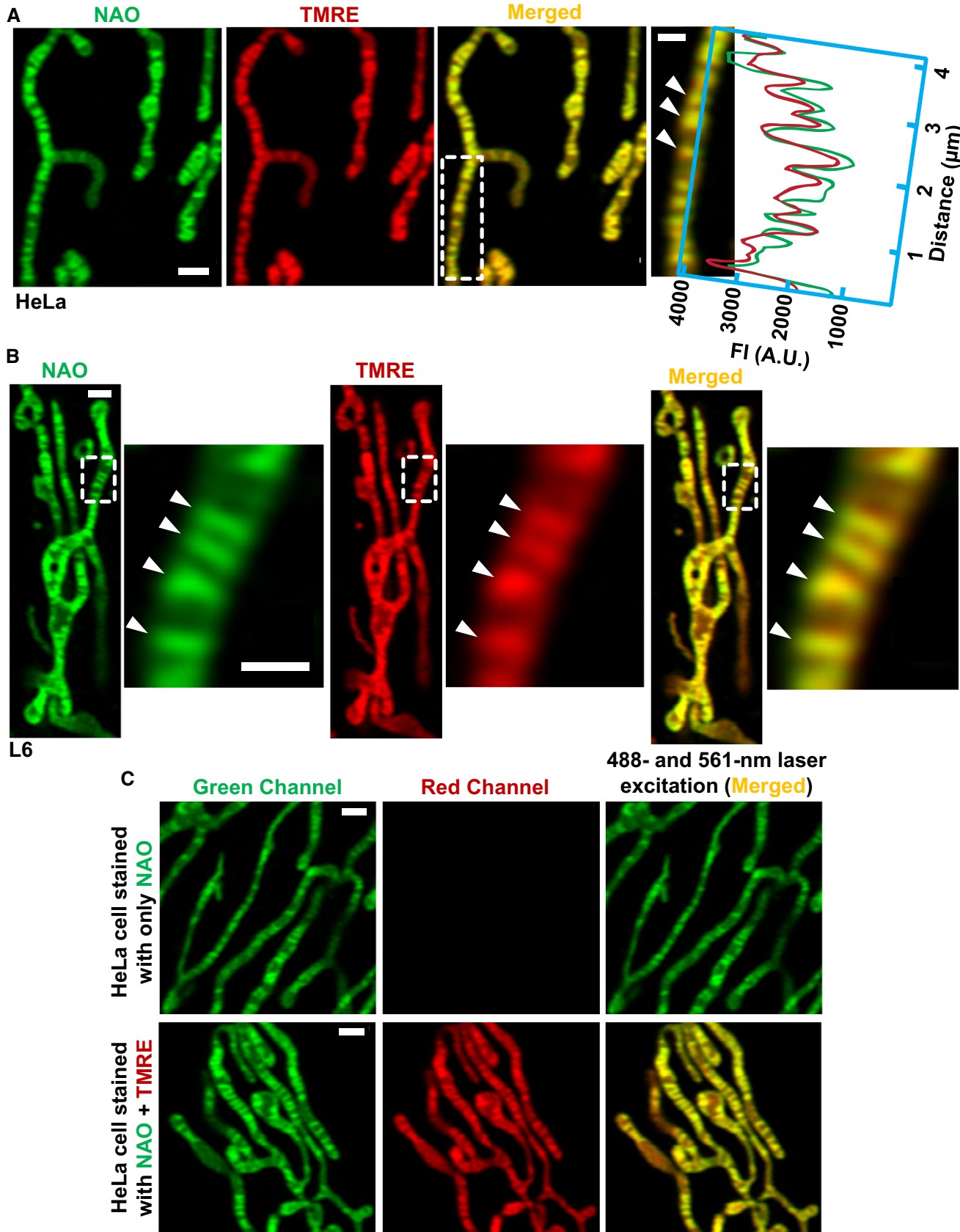

**Figure 2.**

**Figure 2. The $\Delta\Psi_m$-sensitive dye, TMRE, partitions to cristae stained with NAO.**

High-resolution imaging of mitochondria in live cells using the Airyscan module of Zeiss LSM880 confocal microscope.

A   Mitochondria from HeLa cell, co-stained with NAO and TMRE. Area from dashed white box, zoomed to right, shows the red and green intensities of TMRE and NAO colocalized (arrowheads). Scale bars = 500 nm. *N* = 3 independent experiments.

B   Mitochondria in L6 myoblast, co-stained with NAO and TMRE. Area from dashed white box, zoomed to right, shows colocalizing NAO and TMRE at the cristae membrane (arrowheads). Scale bars = 500 nm. *N* = 3 independent experiments.

C   Mitochondria in HeLa cells, stained with NAO alone (top row) vs. NAO + TMRE (bottom row), and simultaneously excited with 488- and 561-nm lasers. Note that, mitochondria stained with NAO alone do not emit noticeable fluorescence in the red channel; only after adding TMRE does strong signal appear in the red channel, showing negligible bleed-through. Scale bars = 500 nm. *N* = 2 independent experiments.

sh-Scramble (sh-Scr) control mitochondria shows structures closely resembling normal cristae (Fig 1E), whereas sh-PTPMT1 mitochondria display a variety of deformed structures (Fig 1F) analogous to cristae perturbations that were previously observed in electron micrographs of PTPMT1-deficient models (Zhang *et al*, 2011). Overall, these data demonstrate that Airyscan technology can resolve mitochondrial ultrastructure in living cells. We subsequently used this approach to measure $\Delta\Psi_m$ at the different compartments along the IMM and determine the level of heterogeneity in $\Delta\Psi_m$ within the individual mitochondrion.

### $\Delta\Psi_m$-dependent dyes colocalize most strongly with cristae

The power-cable model of the $\Delta\Psi_m$ assumes an electrical continuity along the IMM without any electrical resistance (Skulachev, 2001). However, the possibility of heterogeneity in $\Delta\Psi_m$ along the IMM could not be investigated until now. To image $\Delta\Psi_m$ along the IMM, we stained HeLa cells with NAO and TMRE (Farkas *et al*, 1989; Loew *et al*, 1993). Remarkably, TMRE appeared to align with the IMM in a non-homogeneous manner, where the most-intense TMRE signal colocalized with NAO at cristae (Fig 2A). To substantiate this observation, we looked at L6 (rat myoblast) cells, which presented the same heterogeneous pattern as HeLa cells (Fig 2B).

When excited by a 488-nm laser, NAO has an emission spectrum that is limited to green wavelengths. However, in these experiments, we needed to excite NAO with the 488-nm laser while simultaneously exciting TMRE with the 561-nm laser, resulting in NAO being exposed to both 488- and 561-nm lasers. The observed alignment of TMRE signal with the membrane staining by NAO raised the possibility that exciting NAO with the 561-nm laser could result in red light emission and thus be wrongly detected as TMRE. To address this possibility, we imaged cells stained with NAO alone and excited simultaneously with 488- ($NAO_{EX}$) and 561-nm ($TMRE_{EX}$) lasers (Fig 2C, top row). We found that emission of NAO after excitation with the 561-nm laser ($TMRE_{EX}$) was undetectable. After adding TMRE to the cells, initially stained with NAO alone, and then exciting with the 561-nm laser using the same power, we observed the appearance of strong signal in the red channel, with most-intense pixels colocalizing with NAO at cristae (Fig 2C, bottom row).

To further validate our findings with TMRE and NAO, we used two additional dyes, MitoTracker Green (MTG) and Rhodamine123 (Rho123). MTG covalently binds to various proteins embedded in the cristae membrane and, as such, is considered a $\Delta\Psi_m$-independent dye, although its initial sequestration in mitochondria depends on $\Delta\Psi_m$ (Presley *et al*, 2003). Rho123 is a $\Delta\Psi_m$ probe, which partitions to mitochondria in a transient way, indicating changes to $\Delta\Psi_m$ (Ward *et al*, 2000; Duchen, 2004). We found that MTG colocalized with TMRE, showing a similar heterogeneous pattern (Fig EV2A).

Then, we examined the partitioning of Rho123 and found that it shows the most-intense signal associated with cristae (Fig EV2B).

To further verify that the staining patterns of TMRE depend on $\Delta\Psi_m$, we used a previously described method to influence $\Delta\Psi_m$: Continuous exposure of TMRE-stained mitochondria to the 488-nm laser results in robust and rapid depolarization and repolarization, a phenomenon known as flickering (Duchen *et al*, 1998). We reasoned that the smaller portion of TMRE bound to the membrane in a $\Delta\Psi_m$-independent manner would remain during the flickering event and reveal the level of noise. Moreover, if the differences in TMRE fluorescence intensity (FI) between cristae and IBM depend on $\Delta\Psi_m$, we would expect that the differences between the brightest and dimmest pixels would markedly decrease during depolarization. Conversely, following repolarization, we would expect these differences in pixel intensities to return. Figure 3A shows an example of a mitochondrion from a HeLa cell stained with MTG and TMRE (arrowheads), where we initially observed the heterogeneous patterns of TMRE; however, at ~ 9 s, the mitochondrion depolarized, and the heterogeneous staining pattern of TMRE was lost. Notably, at ~ 16 s, this mitochondrion repolarized and exhibited nearly the same TMRE heterogeneity as before the depolarization. Quantification of the changes in $\Delta\Psi_m$ during the flickering phenomenon demonstrates that ~ 85% of TMRE signal was lost during depolarization (Fig 3B). Moreover, the differences between the brightest and dimmest areas (cristae and matrix, respectively) are attenuated during transient depolarization but are reestablished following restoration of $\Delta\Psi_m$. These data support that the heterogeneous staining patterns of the TMRE are due to differences in $\Delta\Psi_m$.

### Quantification of $\Delta\Psi_m$ differences between cristae and IBM

The Nernst equation can be used to quantify $\Delta\Psi_m$ by acquiring the FI of $\Delta\Psi_m$-sensitive probes (e.g., TMRE). The FIs of the probes at different subcellular compartments can be used to extrapolate the differences in concentrations of the probe, which are needed to calculate the difference in $\Delta\Psi_m$ between compartments (Ehrenberg *et al*, 1988; Farkas *et al*, 1989; Loew *et al*, 1993; Wikstrom *et al*, 2007; Twig *et al*, 2008). We used the average TMRE FI of the mitochondria as a reference point to calculate the $\Delta\Psi_m$ of the different compartments, an approach similar to that employed in multi-electrode ECG. We found the voltage at cristae to be significantly higher than at IBM (Fig 4A and C). These data indicate that the hetero-potential along the IMM consists of at least two basic segments—the cristae and the IBM. Representing pixel intensities in pseudo-color as a LookUp Table (LUT), where white and blue correspond to the highest and lowest $\Delta\Psi_m$, respectively, it is apparent that the voltage associated with cristae (arrowheads) is generally higher than IBM,

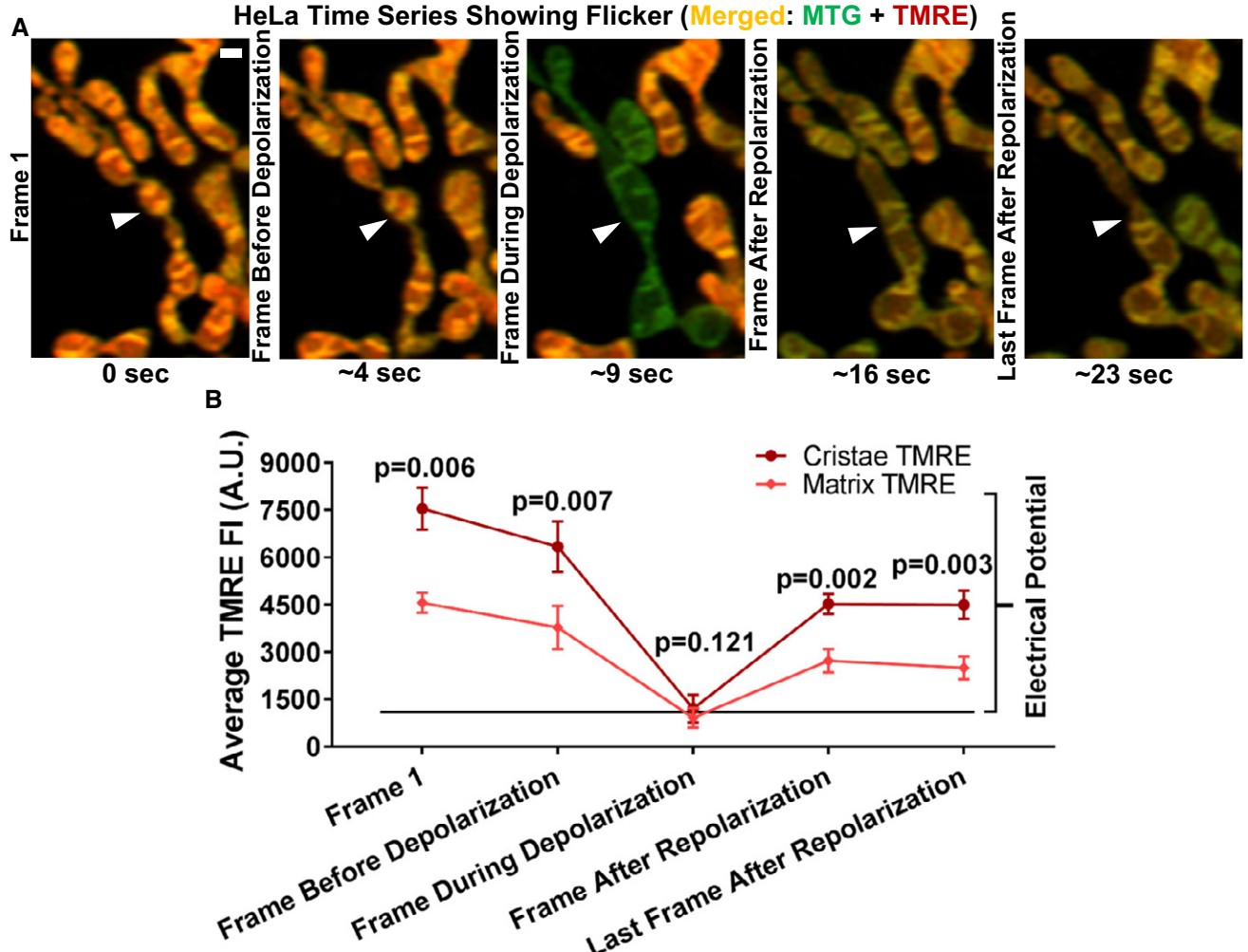

**Figure 3. Validation that TMRE partitioning to cristae is $\Delta\Psi_m$-dependent.**

A   Time-lapse Airyscan imaging of $\Delta\Psi_m$ in living HeLa cell, co-stained with MTG ($\Delta\Psi_m$-insensitive after loading) and TMRE ($\Delta\Psi_m$ sensitive). Arrowhead points to a flickering event where a mitochondrion depolarizes (~ 9 s) and repolarizes (~ 16 s), showing that heterogeneous signal from TMRE (but not MTG) disappears and reappears. Scale bar = 500 nm. $N = 4$ independent experiments.

B   Quantification of (A). Plot shows average TMRE fluorescence intensity (FI) of cristae (dark red line) vs. matrix (light red line) during the time series. The drop in TMRE FI during the depolarization phase of the flickering is the $\Delta\Psi_m$-sensitive component of the TMRE signal. The remaining TMRE FI during depolarization can be considered as the $\Delta\Psi_m$-insensitive portion of TMRE signal. Note that, the remaining TMRE FI after depolarization at the cristae and matrix is approximately identical, indicating that differences in TMRE FI between the cristae and matrix prior to depolarization are derived from differences in $\Delta\Psi_m$. $N = 4$ independent experiments.

Data information: Data were analyzed with 2-tailed Student's *t*-tests, and *P* values < 0.05 were considered statistically significant. Specific *P* values are indicated in the figure. Error bars indicate SEM.

emphasizing the electrochemical discontinuity between these contiguous regions of the IMM (Fig 4B).

We next explored whether the $\Delta\Psi_m$ of primary cells would follow the same pattern. We stained mitochondria in primary hepatocytes with TMRE and found $\Delta\Psi_m$ heterogeneity similar to that in HeLa cells, indicating that discrete electrochemical domains also exist along the IMM of differentiated cells with strong mitochondrial oxidative function (Fig 4D–F).

To further confirm the $\Delta\Psi_m$ differences between cristae and IBM observed using Airyscan imaging, we determined whether the $\Delta\Psi_m$ would display the same heterogeneity by super-resolution microscopy (e.g., STED). Using living HeLa cells stained with TMRM, we found a nearly identical pattern in the heterogeneity of $\Delta\Psi_m$, where, notably, cristae $\Delta\Psi_m$ significantly exceeds that of IBM (Fig 4G–I). Altogether, these data demonstrate that the voltage associated with cristae is significantly higher than that of IBM, which is consistent with the higher concentration of ETC components associated with cristae membranes.

**$\Delta\Psi_m$ differences between cristae and IBM ($\Delta\Psi_{Cr-IBM}$) are sensitive to inhibition of $F_1F_0$ ATP Synthase and to uncoupling**

$F_1F_0$ ATP Synthase is primarily localized to the rims of cristae (Dudkina *et al*, 2005; Strauss *et al*, 2008; Davies *et al*, 2011). If cristae

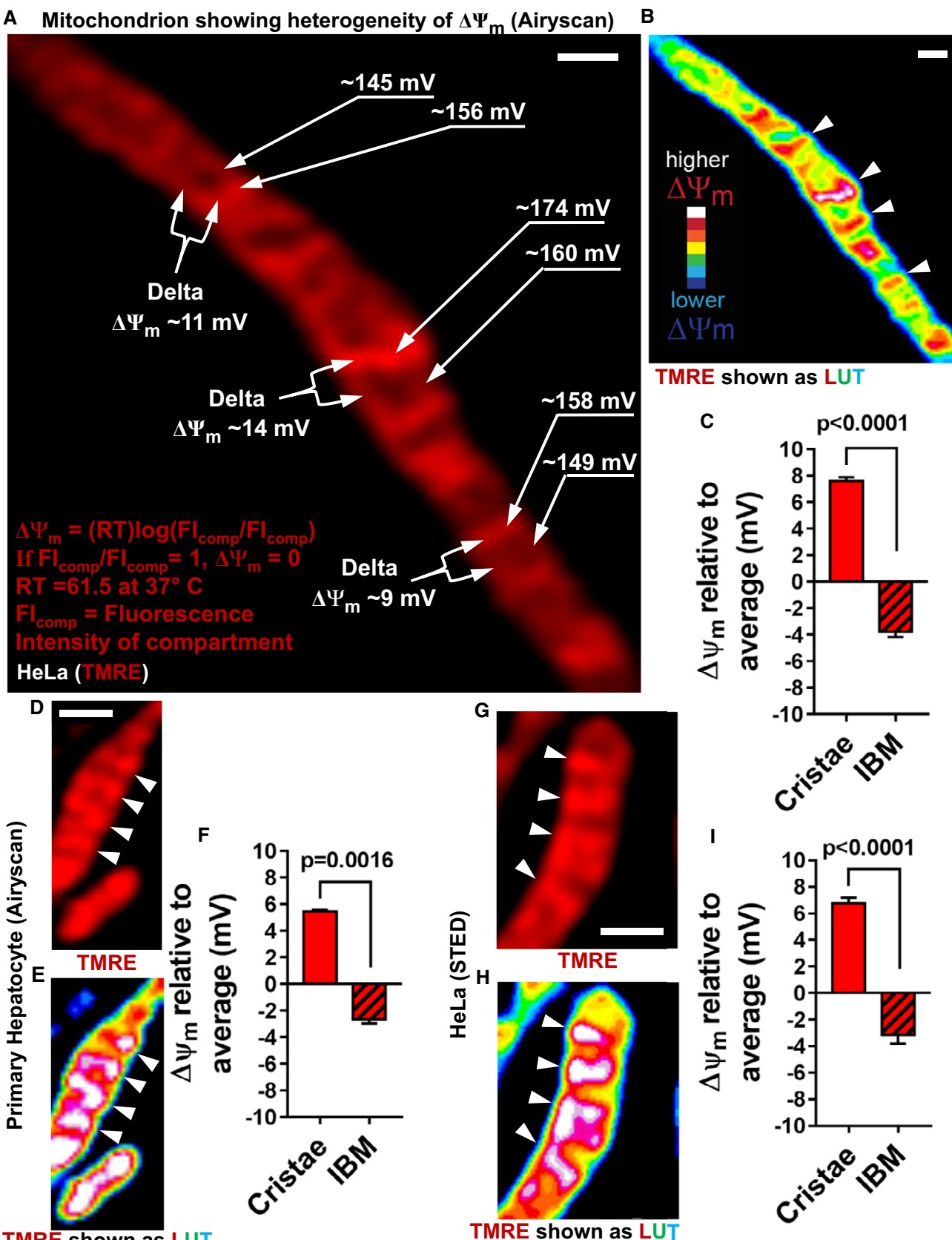

**A** Mitochondrion showing heterogeneity of $\Delta\Psi_m$ (Airyscan)

~145 mV
~156 mV
Delta $\Delta\Psi_m$ ~11 mV
~174 mV
~160 mV
~158 mV
~149 mV
Delta $\Delta\Psi_m$ ~14 mV
Delta $\Delta\Psi_m$ ~9 mV

$\Delta\Psi_m = (RT)log(Fl_{comp}/Fl_{comp})$
If $Fl_{comp}/Fl_{comp}= 1$, $\Delta\Psi_m = 0$
RT =61.5 at 37° C
$Fl_{comp}$ = Fluorescence
Intensity of compartment
HeLa (TMRE)

**B**
higher $\Delta\Psi_m$
lower $\Delta\Psi_m$
TMRE shown as LUT

**C**
p<0.0001
$\Delta\Psi_m$ relative to average (mV)
Cristae    IBM

**D**
Primary Hepatocyte (Airyscan)
TMRE

**E**
TMRE shown as LUT

**F**
p=0.0016
$\Delta\Psi_m$ relative to average (mV)
Cristae    IBM

**G**
TMRE

**H**
HeLa (STED)
TMRE shown as LUT

**I**
p<0.0001
$\Delta\Psi_m$ relative to average (mV)
Cristae    IBM

**Figure 4.**

**Figure 4. Cristae and IBM have different $\Delta\Psi_m$.**

A Live-cell Airyscan image of mitochondrion in HeLa cell, showing different membrane potentials in different mitochondrial regions. Membrane potentials were calculated based on TMRE FI differences between compartments ($FI_{comp}$). Regions of interest from cristae and IBM used for $\Delta\Psi_m$ calculations are labeled with arrows on the right-hand side of the mitochondrion, using the cytosol as the reference compartment. Labeled with forks on the left-hand side of the mitochondrion are the calculations of $\Delta\Psi_m$ between individual cristae and their neighboring IBM. Nernst equation used to calculate different voltages. Scale bar = 500 nm.

B LUT of mitochondrion in HeLa cell shown in (A), color coding of TMRE FIs on scale of white (most intense) to blue (least intense): LUT scale shown in lower left-hand corner. Arrowheads indicate cristae. Scale bar = 500 nm. Note that, most-intense pixels (white) only associate with cristae.

C Quantification of $\Delta\Psi_m$ (mV) at cristae and IBM relative to the average $\Delta\Psi_m$ of the whole mitochondrion, calculated as in (A). Note: $\Delta\Psi_m$ at cristae is significantly higher than $\Delta\Psi_m$ at IBM, indicating electrochemical boundaries separate these two regions of the IMM. $N = 3$ independent experiments.

D–F Live-cell Airyscan imaging of mitochondria in cultured primary mouse hepatocytes stained with TMRE. Arrowheads denote cristae. (D) Image showing TMRE-labeled mitochondria. (E) LUT color coding of TMRE FI. (F) Quantification of $\Delta\Psi_m$ differences using Nernst equation as shown in (A). Scale bar = 500 nm. $N = 2$ independent experiments.

G–I Live-cell STED imaging of mitochondria in HeLa cell stained with TMRM. Arrowheads show cristae. Scale bar = 500 nm. (G) Image showing TMRM-labeled mitochondrion. (H) LUT color coding of TMRM FI. (I) Quantification of $\Delta\Psi_m$ differences using Nernst equation as shown in (A). Scale bar = 500 nm. $N = 4$ independent experiments. Note that, calculation of $\Delta\Psi_m$ differences between the compartments based on images captured by STED and by Airyscan was very similar.

Data information: Data were analyzed with 2-tailed Student's $t$-tests, and $P$ values < 0.05 were considered statistically significant. Specific $P$ values are indicated in the figure. Error bars indicate SEM.

---

maintain $\Delta\Psi_m$ that is distinct from the IBM, then inhibiting the consumption of the proton gradient by $F_1F_0$ ATP Synthase would further increase the difference between IBM and cristae. To assess this hypothesis, we inhibited $F_1F_0$ ATP Synthase with oligomycin and measured the difference in the $\Delta\Psi_m$ between the cristae and IBM ($\Delta\Psi_{Cr\text{-}IBM}$). We found that not only did oligomycin increase the total $\Delta\Psi_m$ (Fig 5A and C), as expected (Farkas *et al*, 1989), but it also increased the $\Delta\Psi_{Cr\text{-}IBM}$, hyperpolarizing the cristae (Fig 5B and C). Conversely, we tested whether treatment with FCCP, a protonophore that shuttles protons across the IMM, would diminish the $\Delta\Psi_{Cr\text{-}IBM}$. Consistent with previous studies (Farkas *et al*, 1989; Loew *et al*, 1993), we determined that FCCP decreased the total $\Delta\Psi_m$ (Fig 5A and C) and resulted in mitochondrial fragmentation (Fig EV3A and B) (Duvezin-Caubet *et al*, 2006; Ishihara *et al*, 2006; Griparic *et al*, 2007). Furthermore, we observed that treatment with FCCP decreased the $\Delta\Psi_{Cr\text{-}IBM}$ (Fig 5B–D).

**The crista-junction (CJ) modulators, MICOS complex and Opa1, regulate $\Delta\Psi_{Cr\text{-}IBM}$**

To determine the role of CJs and cristae architecture in the generation of $\Delta\Psi_{Cr\text{-}IBM}$, we used cellular models in which proteins controlling the formation of cristae and CJs are perturbed. To date, the most notable of these are the family of MICOS-complex proteins (Rabl *et al*, 2009; Harner *et al*, 2011; Hoppins *et al*, 2011; von der

Malsburg *et al*, 2011; Zerbes *et al*, 2012; Barbot *et al*, 2015; Friedman *et al*, 2015; Guarani *et al*, 2015; Barrera *et al*, 2016; Hessenberger *et al*, 2017; Rampelt *et al*, 2017; Wollweber *et al*, 2017) and the GTPase, Opa1 (Frezza *et al*, 2006; Cogliati *et al*, 2016). To determine whether the MICOS complex and/or Opa1 are required to preserve the $\Delta\Psi_{Cr\text{-}IBM}$, we examined $\Delta\Psi_{Cr\text{-}IBM}$ in cells with either Opa1 or MICOS-complex subunits deleted.

Firstly, we examined the effect of deleting Mic13 on $\Delta\Psi_{Cr\text{-}IBM}$ in HeLa cells. While Mic13-KO cells showed a marked decrease in the number of cristae, we identified some regions of IMM that maintained cristae structures (Fig 6A; arrowheads). Mic13-KO mitochondria showed decreased heterogeneity of TMRE staining along the IMM and $\Delta\Psi_{Cr\text{-}IBM}$ was significantly diminished, compared to control HeLa cells (Fig 6A and B). To further address the role of the MICOS complex in regulating the mitochondrial hetero-potential, we examined cells in which Mic60 or Mic10 were deleted (Fig EV1C). In these cells, we observed a significant diminishment of heterogeneity of TMRE FI along the IMM (Fig 6C), as well as a drop in $\Delta\Psi_{Cr\text{-}IBM}$ (Fig 6D and E).

We next studied the effects of Opa1 deletion on $\Delta\Psi_{Cr\text{-}IBM}$. Various studies have demonstrated that not only does Opa1 associate with components of the MICOS complex (Barrera *et al*, 2016; Glytsou *et al*, 2016), but it appears to have an independent function as a molecular staple, holding the cristae in a closed configuration (Frezza *et al*, 2006). We hypothesized, therefore, that deletion of

---

**Figure 5. $\Delta\Psi_m$ differences between cristae and IBM are sensitive to $F_1F_0$ ATP Synthase inhibition and uncoupling.**

Response of $\Delta\Psi_m$ to the $F_1F_0$ ATP Synthase inhibitor, oligomycin (10 μM), compared to the uncoupler, FCCP (10 μM), determined using live-cell Airyscan imaging of HeLa cells co-stained with MTG and TMRE.

A Quantification of total mitochondrial TMRE FI. $N = 3$ independent experiments.

B Quantification of cristae $\Delta\Psi_m$ relative to IBM ($\Delta\Psi_{Cr\text{-}IBM}$) in response to oligomycin compared to FCCP. $F_1F_0$ ATP Synthase is exclusively localized to the cristae and significant increase in $\Delta\Psi_{Cr\text{-}IBM}$ in response to blocking of $F_1F_0$ ATP Synthase with oligomycin indicates that differences in TMRE FI between cristae and IBM are driven by oxidative phosphorylation. $N = 3$ independent experiments.

C Representative Airyscan images showing mitochondria from living HeLa cells stained with MTG and TMRE. Cells were first stained for 1 h and only then treated with oligomycin or FCCP. Images show hyperpolarization of cristae under oligomycin (middle row; arrowheads) and depolarization of cristae under FCCP (bottom row; arrowheads) compared to control (top row; arrowheads). Scale bar = 500 nm. $N = 3$ independent experiments. Note that, the $\Delta\Psi_m$-independent fraction of TMRE staining is < 5% of the signal in control conditions.

D Zoomed-in region of FCCP-treated mitochondria in (C). Increased contrast was used to visualize TMRE to compensate for TMRE loss induced by FCCP. These images show diminished heterogeneity of $\Delta\Psi_{Cr\text{-}IBM}$ values (arrowheads). Scale bar = 500 nm.

Data information: Data were analyzed with 2-tailed Student's $t$-tests, and $P$ values < 0.05 were considered statistically significant. Specific $P$ values are indicated in the figure. Error bars indicate SEM.

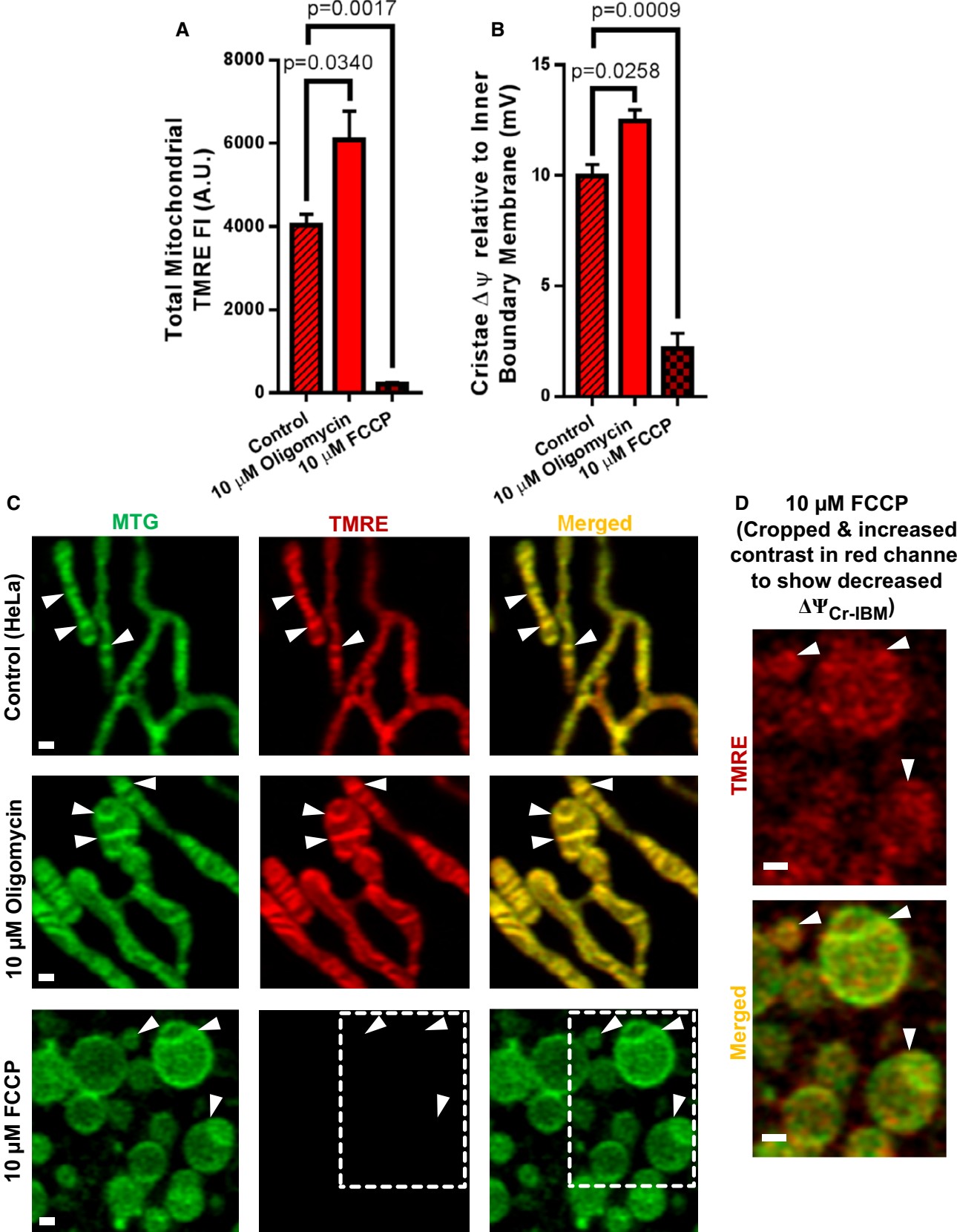

**Figure 5.**

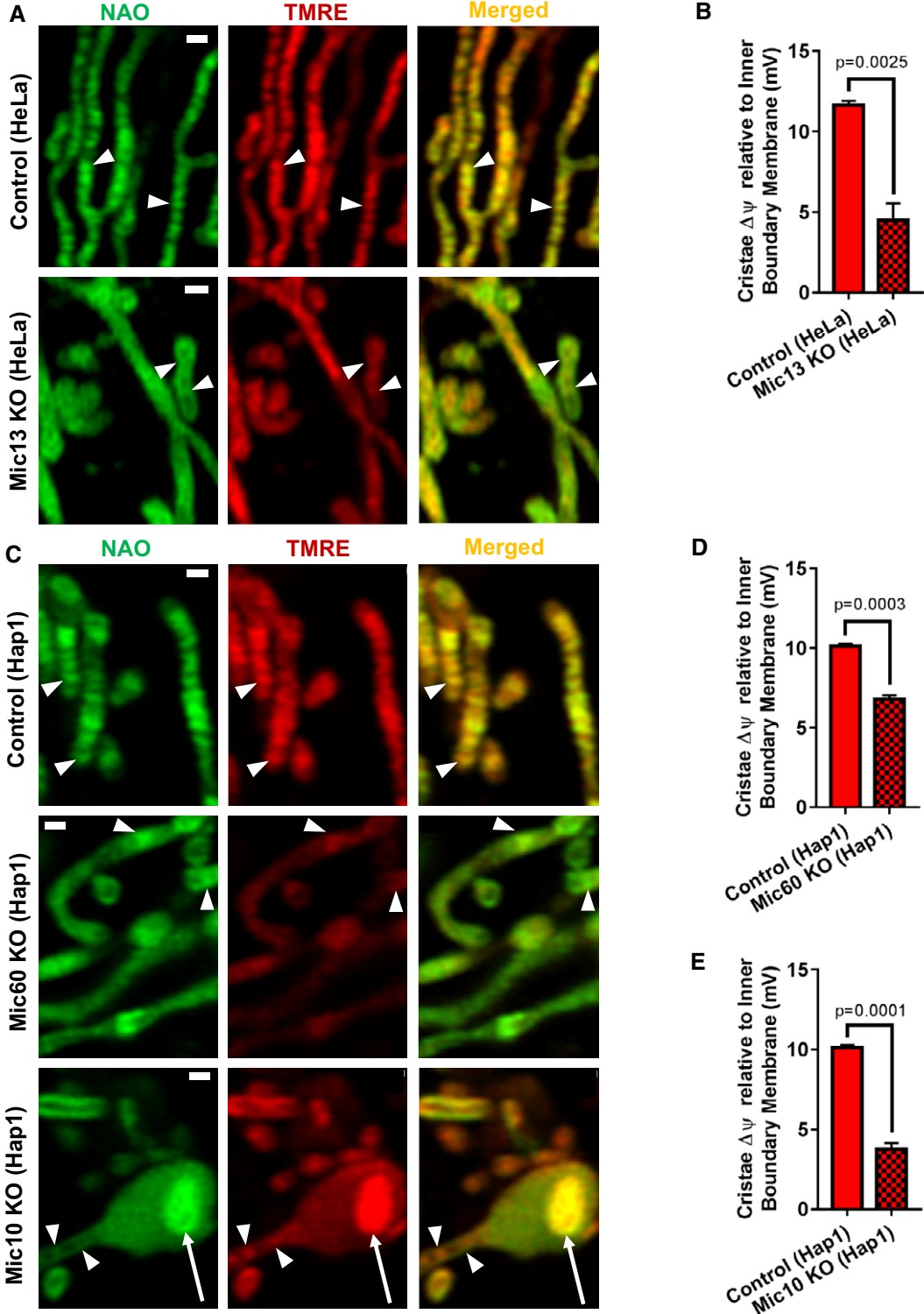

**Figure 6.**

Opa1 (Fig EV1D), like deletion of components of the MICOS complex, would disrupt the electrochemical boundaries between the cristae and the IBM, thereby equilibrating the potential along the IMM. Our data indicate that deletion of Opa1 indeed results in a significant loss in $\Delta\Psi_{Cr-IBM}$ (Fig 7A and B). Altogether, these data indicate that cristae architecture and CJs are essential for the formation of the electrochemical boundaries that allow for $\Delta\Psi_m$ of cristae to remain different from that of IBM.

**Figure 6. Crista junctions (CJs) regulate the difference in $\Delta\Psi_m$ between cristae and IBM.**

The role of cristae structure as well as CJ formation and sealing on the generation of the difference in $\Delta\Psi_m$ between cristae and IBM were studied by disrupting cristae using Mic10-, Mic13-, and Mic60-deficient cells. Mic13 and Mic60 support cristae formation. Mic10 is essential for CJ formation, and, in its absence, cristae tend to remain as vesicles detached from IBM. Live-cell Airyscan imaging of TMRE was used in all figure panels and models.

A  Representative images of mitochondria in control (top row) vs. Mic13-KO (bottom row) HeLa cells. Note that, TMRE FI in Mic13-KO mitochondria is distributed more evenly along the IMM, so that $\Delta\Psi_{Cr-IBM}$ is decreased (arrowheads). Scale bar = 500 nm. $N$ = 3 independent experiments.

B  Quantification of (A), showing deletion of Mic13 in HeLa cells results in significant decrease in $\Delta\Psi_{Cr-IBM}$. While Mic13-KO cells show a substantial decrease in cristae number, the loss of cristae does not appear absolute, making it feasible to compare TMRE FI at cristae relative to IBM. $N$ = 3 independent experiments.

C  Representative images of Hap1 control (top row) vs. Mic60 KO (middle row) and Mic10 KO (bottom row); Mic60-KO mitochondria show very few cristae structures. TMRE staining along the IMM is relatively homogeneous, indicating decreased $\Delta\Psi_{Cr-IBM}$. In general, Mic10 KO in Hap1 cells results in decreased TMRE signal intensity at cristae relative to IBM (arrowheads) as compared to control cells. Deletion of Mic10 induces detachment of cristae from IBM (arrows; see Fig 8). Except for the detached cristae vesicles, the TMRE staining is homogeneously distributed in Mic10-KO cells, indicating decreased $\Delta\Psi_{Cr-IBM}$. Scale bar = 500 nm. $N$ = 3 independent experiments.

D  Quantification of (C), showing Mic60 KO in Hap1 cells results in significant decrease in $\Delta\Psi_{Cr-IBM}$. While Mic60-KO cells show a marked loss of normal cristae structures, depletion of cristae does not appear absolute, making it possible to compare TMRE FI at cristae relative to IBM. $N$ = 3 independent experiments.

E  Quantification of (C), showing Mic10 KO in Hap1 cells results in a significant decrease in $\Delta\Psi_{Cr-IBM}$. Note: quantification of $\Delta\Psi_m$ at cristae relative to IBM was performed only on cristae structures that appeared to maintain attachment to IBM. $N$ = 3 independent experiments.

Data information: Data were analyzed with 2-tailed Student's $t$-tests, and $P$ values < 0.05 were considered statistically significant. Specific $P$ values are indicated in the figure. Error bars indicate SEM.

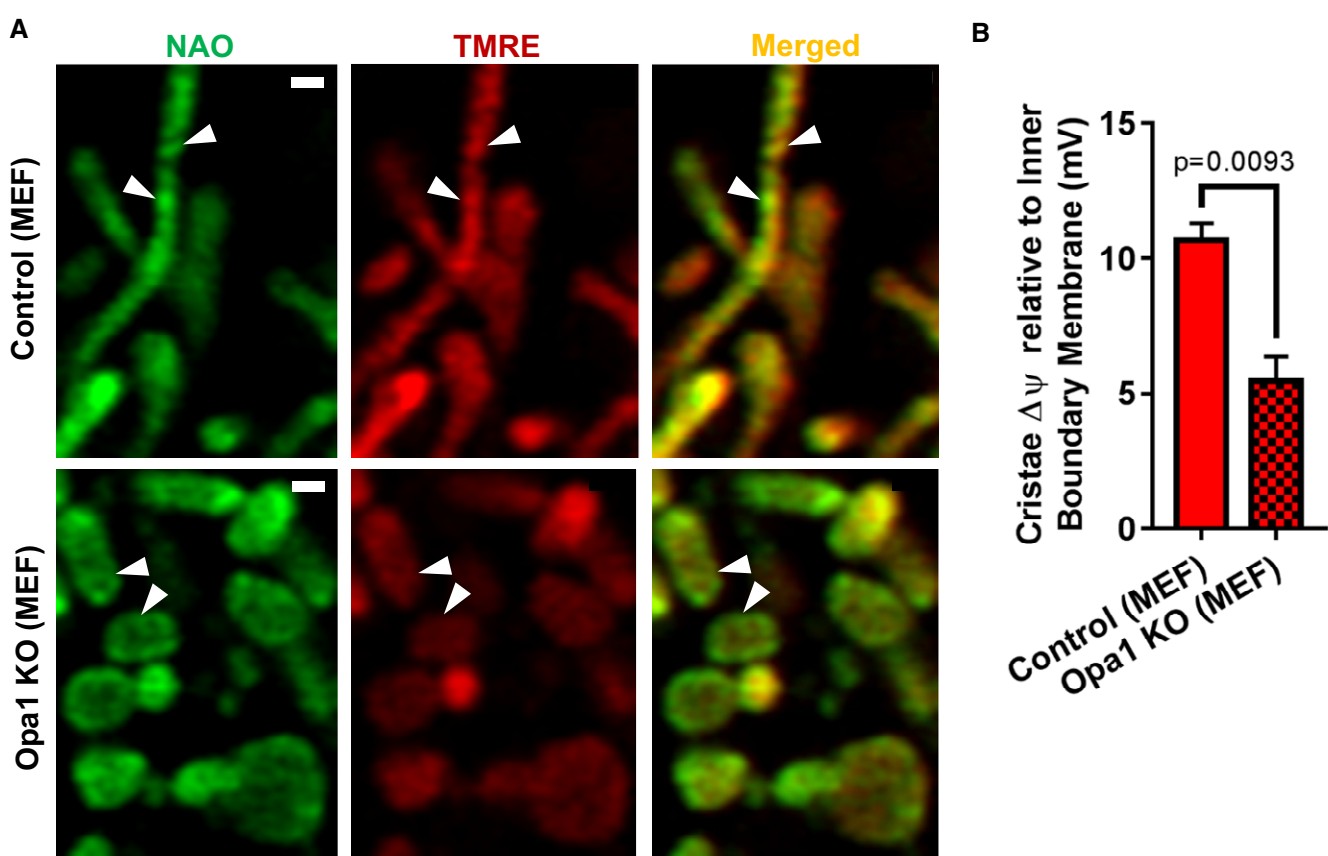

**Figure 7. Opa1 regulates the difference in $\Delta\Psi_m$ between cristae and IBM.**

Studies show that Opa1 interacts with MICOS complex, promoting closure of CJs. Thus, we tested whether Opa1 was required to maintain the difference in $\Delta\Psi_m$ between cristae and IBM.

A  Live-cell Airyscan images of MEF control (top row) vs. Opa1 KO (bottom row), co-stained with NAO and TMRE. Arrowheads indicate decreased intensity of TMRE FI at cristae compared to IBM. Note the more even distribution of TMRE staining, indicating the cristae and IBM are relatively equipotential. Scale bar = 500 nm.

B  Quantification of (A), showing Opa1-KO MEFs have significantly decreased $\Delta\Psi_{Cr-IBM}$. $N$ = 3 independent experiments.

Data information: Data were analyzed with 2-tailed Student's $t$-tests, and $P$ values < 0.05 were considered statistically significant. Specific $P$ values are indicated in the figure. Error bars indicate SEM.

## A  Laser-induced Depolarization Time Series (TMRE)

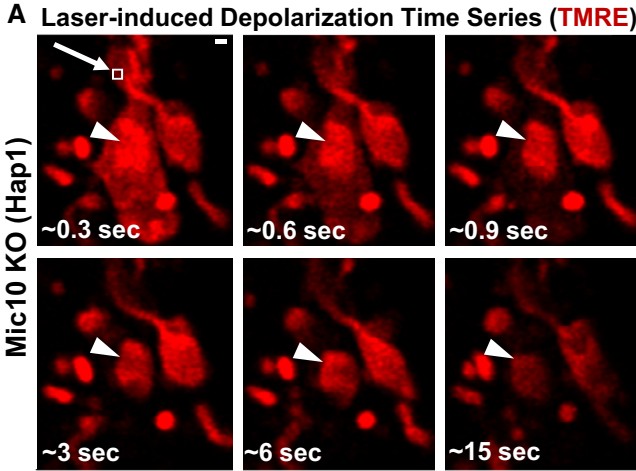

## B

### TMRE

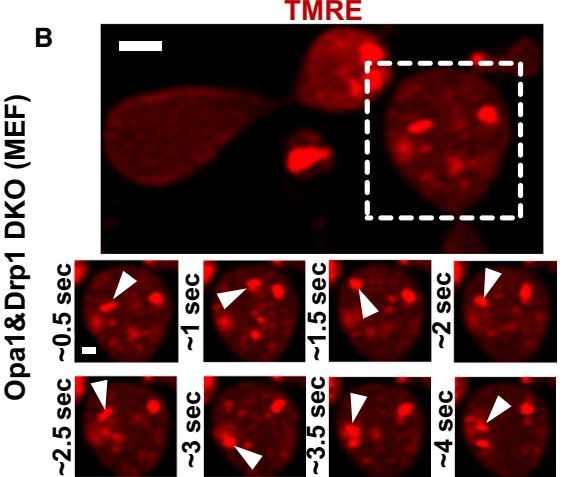

**Figure 8.  The largest differences in $\Delta\Psi_{Cr\text{-}IBM}$ arise in cristae vesicles generated by deleting CJ regulators.**

A  Live-cell Airyscan images of laser-induced depolarization time series of Mic10 KO Hap1 cell stained with TMRE. Mitochondrion exposed to rapid, high 2-photon laser power in region of white box, arrow (~ 0.3 s); large, hyperpolarized vesicle (arrowhead) remains polarized after the rest of the mitochondrion loses $\Delta\Psi_m$, indicating electrochemical discontinuity between hyperpolarized vesicles and the rest of the mitochondrion, including the IBM. Scale bar = 500 nm. $N$ = 2 independent experiments.

B  Live-cell Airyscan image of time series showing hyperpolarized vesicles (arrowheads; region cropped from dashed white box) inside mitochondria lacking both Opa1 and Drp1. Scale bar = 500 nm. $N$ = 2 independent experiments. Free movement of hyperpolarized vesicles within the matrix indicate their detachment from the IBM, representing an extreme case of cristae independency within one mitochondrion (see Movie EV1 for time series).

Data information: For quantification of hyperpolarized vesicles, see values in main text.

## The largest differences in $\Delta\Psi_{Cr\text{-}IBM}$ arise in cristae vesicles, generated by deleting CJ regulators

Intriguingly, in ~ 25% of Mic10 KO cells, we observed structures that appeared to have detached from the IBM and become hyperpolarized (Fig 6C, lowest row, arrow). To test whether these

hyperpolarized structures in Mic10 KO cells had no membrane continuity with the IMM, we performed laser-induced depolarization of mitochondria that included such vesicles. If these hyperpolarized vesicles were unanchored from the IMM, we would expect that they would remain polarized despite the collapse of $\Delta\Psi_m$ elsewhere in the mitochondrion. Figure 8A shows that such a hyperpolarized structure (arrowhead) maintained its $\Delta\Psi_{vesicle}$ for at least 15 s after the rest of the mitochondrion had depolarized. We determined that the $\Delta\Psi_{vesicle\text{-}IBM}$ of such vesicles was significantly higher than cristae that maintained attachment to the IBM ($\Delta\Psi_{Cr\text{-}IBM}$ for Control cristae = 10.11 mV vs. $\Delta\Psi_{vesicle\text{-}IBM}$ for Mic10 KO vesicles = 25.85 mV; $P$ = 0.0005; $N$ = 3 independent experiments).

We further hypothesized that deletion of both Opa1 and Drp1 (Fig EV1D) would result in an extreme case of electrochemical discontinuity of the IMM. Indeed, our imaging of Opa1&Drp1 DKO MEFs demonstrates the formation of strongly hyperpolarized vesicles moving haphazardly within the matrix (Fig 8B; Movie EV1 shows time series of hyperpolarized vesicles). Compared to control MEFs that had a $\Delta\Psi_{Cr\text{-}IBM}$ of 10.67 mV, the hyperpolarized vesicles in the Opa1&Drp1 DKO MEFs showed a $\Delta\Psi_{vesicle\text{-}IBM}$ of 39.55 mV ($P$ = 0.0014; $N$ = 3 independent experiments).

Hyperpolarized vesicles were quite rare in control cells, suggesting that they are formed in response to bioenergetic dysfunction and/or impaired cristae formation.

## Different cristae within a mitochondrion can have different $\Delta\Psi_m$, indicating that cristae function as independent bioenergetic units

Previous studies suggest that a single mitochondrion constitutes a single bioenergetic unit (Amchenkova et al, 1988; Skulachev, 2001; Glancy et al, 2015). A typical experiment would involve laser-induced depolarization at one tip of a mitochondrion, which would appear to instantaneously result in a complete collapse of the $\Delta\Psi_m$, leading to the conclusion that the entire organelle functions as one electrochemical unit. These time-lapse experiments determined $\Delta\Psi_m$ at intervals of 5 s or longer after laser-induced depolarization and were employing microscopes that allowed for single mitochondrion, rather single crista, resolution (Amchenkova et al, 1988; Skulachev, 2001). A 5-s interval, however, is enough time to generate and propagate a soluble signaling molecule and/or changes in membrane architecture that could induce depolarization of independent cristae structures in an entire mitochondrion. Thus, to test whether laser-induced depolarization reflected (i) an instantaneous collapse in voltage due to the mitochondrion functioning as an uninterrupted electrochemical conduit; or (ii) a gradual loss of voltage due to a chemical signal and/or structural changes propagating through the organelle, we re-ran the same experiments as previously described but acquired images ~ 20× more rapidly. Visualizing mitochondria from rat myoblasts (L6) using Rho123 (Fig 9A; Movie EV2 shows time series), we exposed a small mitochondrial region (~ 0.5–1 $\mu m^2$) to a rapid, high-power pulse of the 2-photon (2-P) laser (arrow, white box). We then observed mitochondria depolarizing in a wavelike (i.e., gradual) manner: The $\Delta\Psi_m$-dependent dye dispersed more rapidly near the original site of laser stress, prior to dispersing along areas farther away (arrowheads follow the loss of $\Delta\Psi_m$ over time). The Rho123 signal is displayed as a multicolored LUT, where pixel intensity is color-coded on a scale where the most-

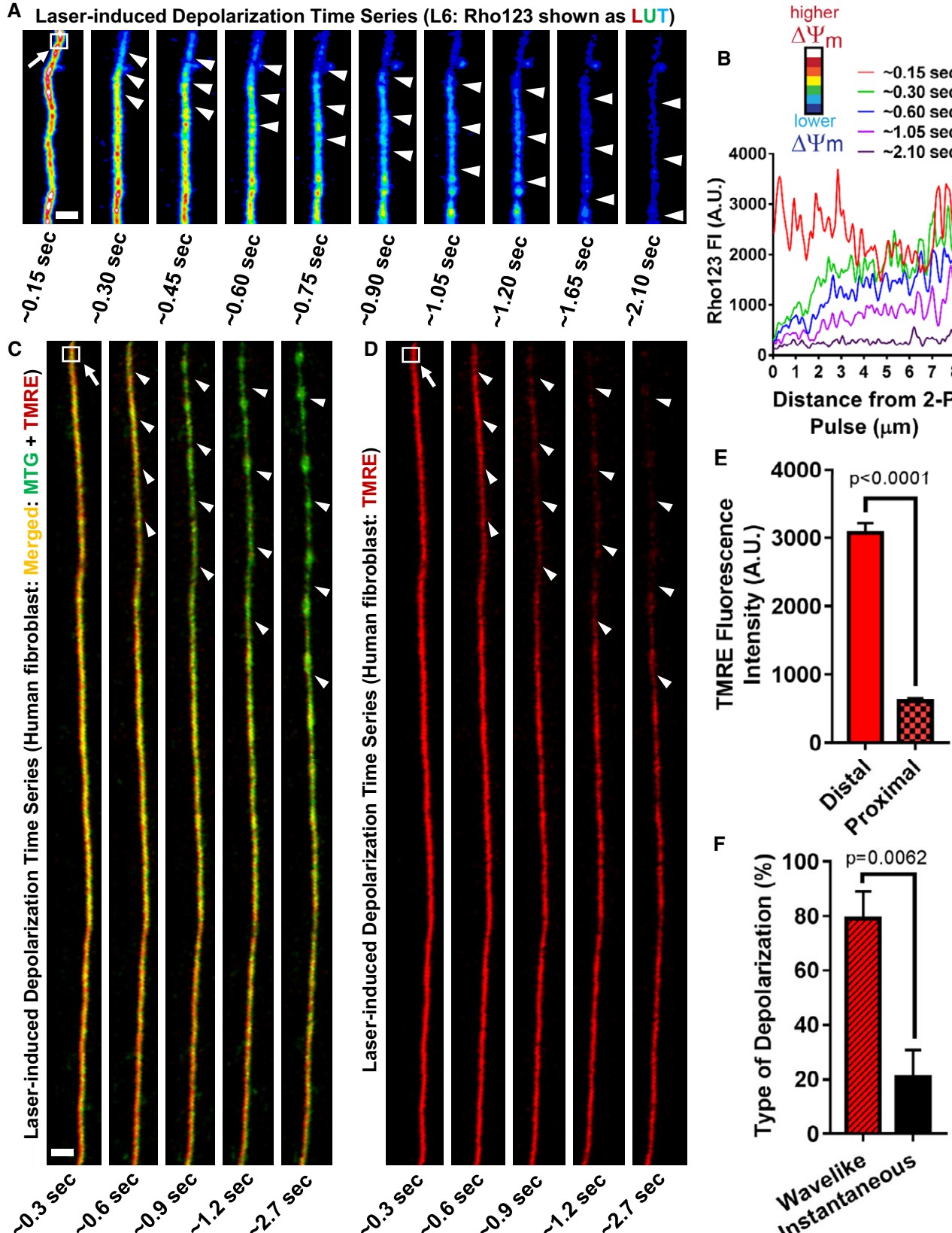

**Figure 9.**

◀

**Figure 9.　Laser-induced depolarization results in non-instantaneous loss of $\Delta\Psi_m$ along the single mitochondrion.**

Laser-induced mitochondrial membrane depolarization in L6 myoblasts shows gradual depolarization of elements along the IMM, visualized using live-cell Airyscan microscopy.

A　Representative images of laser-induced depolarization time series from L6 myoblast, stained with $\Delta\Psi_m$-dependent dye, Rho123. Images show LUT color-coded for Rho123 FI. White and blue pixels represent most- and least-intense $\Delta\Psi_m$, respectively. See legend of LUT colors on the right. Note that, at the first time point (~ 0.15 s), white arrow points to box marking area of mitochondrion exposed to phototoxic pulse of 2-photon laser (≤ 5 ms), inducing depolarization. Ensuing frames show wavelike depolarization away from site of perturbation (arrowheads). Scale bar = 500 nm. *N* = 3 independent experiments. (See Movie EV2 for time series.)

B　Profile plot showing Rho123 pixel intensity as function of distance along mitochondrion in (A). Note that, zero value in the *x*-axis corresponds to top area of the mitochondrion shown in (A), near site of 2-photon laser exposure (box). Red line corresponds to frame immediately before depolarization (~ 0.15 s); green, blue, purple, and dark purple lines correspond to ~ 0.30, ~ 0.60, ~ 1.05, and ~ 2.10 s, respectively, after initial depolarization, showing wavelike dissipation of $\Delta\Psi_m$ over time.

C–F　Laser-induced mitochondrial membrane depolarization in living human skin fibroblasts co-stained with TMRE and MTG and imaged with Airyscan, showing gradual depolarization of elements along the IMM. (C) Time-lapse images of merged green (MTG) and red (TMRE) channels following laser-induced mitochondrial membrane depolarization. Note that, at the first time point (0.3 s), white arrow points to box marking area of mitochondrion exposed to phototoxic pulse of 2-photon laser (≤ 5 ms), inducing depolarization. Depolarization is marked by the dissipation of the $\Delta\Psi_m$-dependent dye (TMRE) while the green (MTG) signal persists (arrowheads). Scale bar = 1 μm. *N* = 4 independent experiments. (D) TMRE channel from (C), showing gradual loss of TMRE along the IMM (arrowheads). Note that, although depolarization has largely completed at the area near the phototoxic stimulus already at 0.9 s, some specific regions at the very top maintain $\Delta\Psi_m$ while others depolarize. (E) Quantification of $\Delta\Psi_m$ using TMRE FI from (C, D). Measurement of TMRE pixel intensities immediately after laser-induced depolarization at site distal (≥ 10 μm) vs. proximal (≤ 1 μm) to box. *N* = 4 independent experiments; see specific *P* values in panel. (F) Quantification of (C, D). Percentage of mitochondria that depolarize in a wavelike (i.e., gradual) vs. instantaneous manner after laser-induced depolarization. Note: imaging at high temporal resolution (~ 100–500 ms/frame) reveals wavelike depolarizations predominate, suggesting the $\Delta\Psi_m$ is composed of multiple, disparate electrochemical domains along the IMM. The time scale of propagation of depolarization is slower than the propagation of electrical phenomena. *N* = 4 independent experiments.

Data information: Data were analyzed with 2-tailed Student's *t*-tests, and *P* values < 0.05 were considered statistically significant. Specific *P* values are indicated in the figure. Error bars indicate SEM.

intense pixels appear white (highest $\Delta\Psi_m$) and the least-intense pixels appear blue (lowest $\Delta\Psi_m$). Examining the images immediately following laser-induced depolarization (~ 0.30, ~ 0.45, and ~ 0.60 s), we found that the mitochondrion loses $\Delta\Psi_m$ first in proximity to the site of laser stress, because the pixels become more green and blue, whereas the distal site remains polarized, as indicated by the white and red pixels. In Fig 9B, we plotted the pixel intensity of the mitochondrion in 9A as a function of distance from the top to the bottom at different time points. In the image immediately preceding laser-induced depolarization (red line), the peaks and valleys occupy a similar range of intensities from one end of the mitochondrion to the other. Following depolarization induced by the 2-P laser pulse, Rho123 intensity proximal to the site of laser stress drops substantially, while the signal intensity at the distal end remains almost unchanged (green line). These asymmetrical changes in FI are visible at later time points (blue and purple lines), until ~ 2.1 s, when the mitochondrion has lost virtually all of its $\Delta\Psi_m$ (dark purple line).

In previous studies, laser-induced depolarization experiments were often performed on mitochondria in fibroblasts, which tend to be remarkably elongated. Therefore, fibroblasts represent an optimal system to test the model of the mitochondrion as an "electrical wire" or "power cable", where loss of $\Delta\Psi_m$ in one place would lead to simultaneous depolarization across the whole organelle (Amchenkova *et al*, 1988; Skulachev, 2001). Staining mitochondria in human fibroblasts with MTG and TMRE, we performed laser-induced depolarization experiments (Fig 9C). Loading of MTG depends on $\Delta\Psi_m$, but it is retained in mitochondria even after depolarization, because, once inside the organelle, it binds covalently to thiol moieties of various proteins (Presley *et al*, 2003). Partitioning of TMRE to mitochondria, on the other hand, depends entirely on $\Delta\Psi_m$. Following 2-P laser pulsation (arrow, white box), there is greater and greater loss of TMRE over time, while the signal from MTG remains. Figure 9D, which shows the TMRE channel alone, demonstrates that, even though the mitochondrion begins to

depolarize proximal to the site of the initial laser stress, it maintains its $\Delta\Psi_m$, distally, for a period of time. To quantify this asymmetry in $\Delta\Psi_m$, we measured TMRE FI immediately after 2-P laser pulsation at sites distal to the initial perturbation (≥ 10 μm from white box) vs. proximal (≤ 1 μm from white box; Fig 9E).

Quantification shows that in the seconds following laser-induced depolarization, mitochondria tend to remain significantly more polarized at sites distant from targeted laser stress, compared to sites close to it. The time gap of seconds for the propagation of the depolarization wave does not support the notion that the IMM functions as a single electrochemical conduit. We next determined the frequency that mitochondria appeared to depolarize in a wavelike vs. instantaneous manner. Time-lapse imaging at approximately 300 ms/frame, we were significantly more likely to observe wavelike instead of instantaneous depolarization (Fig 9F).

Together, using different cell lines and mitochondrial dyes, our data suggest that the $\Delta\Psi_m$ is organized into multiple, disparate electrochemical domains along the length of the IMM. It further suggests that the laser-induced depolarization phenomenon propagates within the mitochondrion through the diffusion of a signal and/or structural changes rather than by the membrane acting as a continuous electrochemical element.

If cristae possess a measure of functional autonomy, then it is possible that different cristae within a single mitochondrion could maintain significantly different membrane potentials ($\Delta\Psi_{Cr}$) from each other. To determine whether one crista within a single mitochondrion has higher $\Delta\Psi_{Cr}$ compared to a neighboring crista, we first defined the level of stability of $\Delta\Psi_{Cr}$ of individual cristae and then asked if the difference between two cristae is larger than the $\Delta\Psi_{Cr}$ fluctuation observed over time in a single crista. To explore this possibility, we performed time-lapse imaging of mitochondria stained with MTG and TMRE, and we examined the stability of $\Delta\Psi_{Cr}$ generated by different cristae over time. The membrane potential of each crista was calculated by referencing it to the IBM to produce a value of $\Delta\Psi_{Cr\text{-}IBM}$. Figure 10A shows an example of four

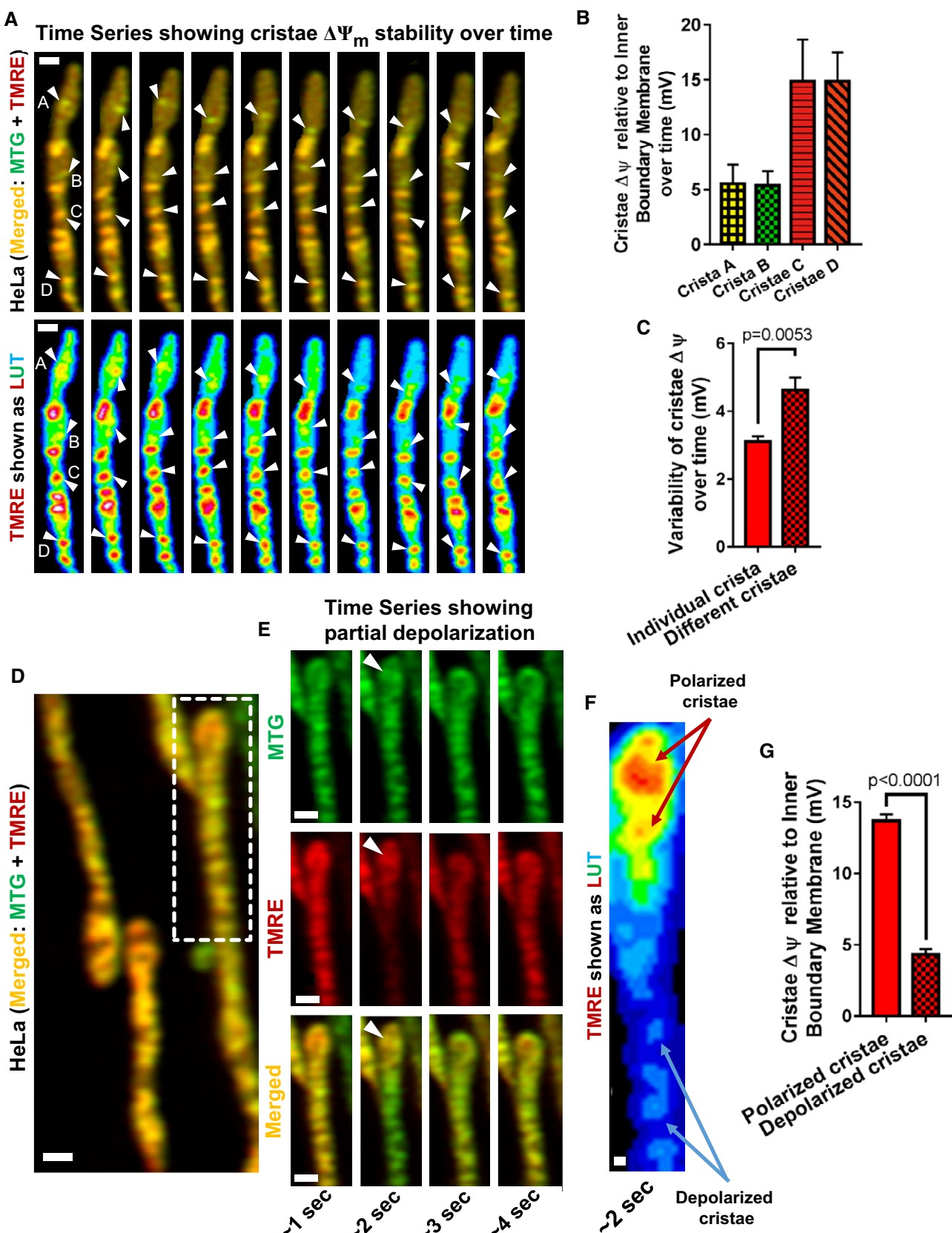

**Figure 10.**

◀

**Figure 10. Cristae are unique electrochemical domains, thus constituting independent bioenergetic units.**

We calculated the $\Delta\Psi_{Cr-IBM}$ between neighboring cristae over time as a surrogate measure of cristae individuality and functional independence, using live-cell Airyscan imaging.

A   Stability of cristae $\Delta\Psi_{Cr-IBM}$ over time. Representative images from time series of HeLa cells co-stained with MTG and TMRE. Top sequence shows merged green (MTG) and red (TMRE) channels. Bottom sequence shows color-coded LUT of TMRE signal of the same sequence shown above. White arrowheads in the image point to specific cristae that were measured over time. Scale bar = 500 nm. $N$ = 5 independent experiments.

B   Quantification of (A), showing average $\Delta\Psi_{Cr-IBM}$ of Cristae "A–D" over time. Note that, Cristae (C) and (D) maintain more than double the $\Delta\Psi_{Cr-IBM}$ of Cristae (A) and (B). Error bars indicate STDEV.

C   Quantification of (A), showing standard deviation in $\Delta\Psi_{Cr-IBM}$ (mV) of individual cristae over time compared to the standard deviation in $\Delta\Psi_{Cr-IBM}$ (mV) among different cristae along a single mitochondrion. $N$ = 5 independent experiments. Note that, there is less variability in $\Delta\Psi_{Cr-IBM}$ of a single crista over time within the same mitochondrion compared to the variability of $\Delta\Psi_{Cr-IBM}$ among cristae over time, indicating cristae possess unique bioenergetic properties.

D–G   Airyscan time series of TMRE flickering in HeLa cells co-stained with MTG and TMRE, showing that depolarization is uncoordinated along the IMM with some cristae depolarizing while neighboring cristae remain polarized. (D) Representative live-cell Airyscan image from time series showing merged green (MTG) and red (TMRE) channels. Area marked by dashed box is shown in the time series in (E). Scale bar = 500 nm. (E) Time series of cropped region from (D), showing tip of mitochondrion (arrowhead) maintaining $\Delta\Psi_m$ while neighboring region of same mitochondrion depolarizes. The capacity of some cristae to maintain polarity, while adjacent cristae transiently depolarize, indicates cristae can exhibit a significant amount of electrochemical autonomy. Scale bar = 500 nm. $N$ = 3 independent experiments. (See Movies EV3 and EV4 for time series). (F) Zoomed-in LUT of mitochondrion from (E) (~ 2 s), showing some cristae remaining polarized (red) while adjacent cristae depolarize (blue). (See Movie EV5 for time series). Scale bar = 100 nm. (G) Quantification of (D–F), showing that during a flickering event, some cristae maintain their membrane potentials, while other cristae depolarize, resulting in large differences in $\Delta\Psi_{Cr-IBM}$ along the individual mitochondrion. $N$ = 3 independent experiments.

different cristae—"A–D" (arrowheads)—that we followed for a period of ~ 10 s. Figure 10B shows that the average $\Delta\Psi_{Cr-IBM}$ over time of Cristae C and D is more than double that of Cristae A and B, indicating that cristae maintain different membrane potentials. To further examine their relative functional stability over time, we quantified the standard deviation of $\Delta\Psi_{Cr-IBM}$ within each crista over time and compared it to the standard deviation of $\Delta\Psi_{Cr-IBM}$ between cristae over time. Our data indicate that, over a period of ~ 10 s, there are significantly larger differences between cristae within a single mitochondrion than the variability recorded from any individual crista over time (Fig 10C). To further investigate the extent to which cristae could exhibit functional autonomy, we studied time-lapse images of mitochondria stained with MTG and TMRE and discovered instances where mitochondria would undergo partial depolarization (Fig 10D–F; arrowhead, arrows; Movies EV3–EV5 show time series of partial depolarization from E, F). Remarkably, instances of partial depolarization of a single mitochondrion revealed cristae that remained polarized despite the depolarization of adjacent cristae. Analyzing the specific frames showing partial depolarization, we quantified the $\Delta\Psi_{Cr-IBM}$ of the polarized cristae vs. depolarized cristae and determined that the polarized cristae maintained a higher $\Delta\Psi_{Cr-IBM}$ than neighboring cristae that appeared to have depolarized (Fig 10G). Altogether, our data demonstrate that cristae display a degree of functional autonomy, highlighting a new role of these critical structures as independent bioenergetic units.

# Discussion

$\Delta\Psi_m$ is the main driving force for proton re-entry through $F_1F_0$ ATP Synthase into the mitochondrial matrix (Mitchell, 1961; Mitchell & Moyle, 1969). In this study, we provide evidence, for the first time, that cristae maintain higher $\Delta\Psi_m$ than IBM and that each individual crista within the same mitochondrion can maintain $\Delta\Psi_m$ distinct from neighboring cristae. Using the LSM880 with Airyscan and STED microscopy, we directly visualized the relationship of the $\Delta\Psi_m$ to the IMM in respiring mitochondria. Our observation that

cristae and IBM have heterogeneous $\Delta\Psi_m$ is based on differential partitioning of $\Delta\Psi_m$-dependent probes to the different segments of the IMM. We provide several lines of evidence to support that this differential partitioning of the $\Delta\Psi_m$-dependent probes is reflecting $\Delta\Psi_m$ and not an artifact of dye binding to membranes: (i) We repeated the observation with the unique $\Delta\Psi_m$-dependent probes, TMRE, TMRM, and Rho123 (Farkas et al, 1989; Loew et al, 1993; Duchen, 2004). If partitioning to cristae were based on non-Nernstian binding, it is unlikely to occur with different dyes. (ii) $\Delta\Psi_{Cr-IBM}$ increased in response to oligomycin and decreased in response to FCCP. (iii) Partitioning to the cristae was decreased in models lacking MICOS-complex subunits or Opa1. (iv) Analyzing flickering events, heterogeneity of TMRE FI along the mitochondrion was markedly decreased during instances of depolarization and reestablished following restoration of $\Delta\Psi_m$. This confirms that TMRE signal stemming from $\Delta\Psi_m$-independent binding of TMRE to the cristae in our experimental system was negligible. (v) The data relating to $\Delta\Psi_{Cr-IBM}$ obtained by Airyscan technology were confirmed by STED microscopy. The imaging was performed in different laboratories with different batches of HeLa cells. (vi) We provided data to confirm the same observations in MEFs, HeLa cells, Hap1 cells, L6 myoblasts, H1975 cells, primary mouse hepatocytes, and human fibroblasts. Altogether, this evidence strongly supports the conclusion that cristae and IBM are sufficiently separated, thus being electrochemically insulated from each other, to allow for different membrane potentials to co-exist along the IMM. A limitation of this methodology is the lack of dyes that are completely independent of $\Delta\Psi_m$ or cardiolipin content. As a result, the NAO and MTG staining of IBM appeared weaker than that of cristae, precluding ratiometric imaging. This limitation only pertains to the differences in $\Delta\Psi_m$ of cristae vs. IBM but would not affect conclusions related to the $\Delta\Psi_m$ differences between different cristae within the same mitochondrion. Moreover, the six points listed above support that the TMRE FI differences we observed between cristae and IBM indeed reflect $\Delta\Psi_m$ differences.

After confirming that the heterogeneous partitioning of $\Delta\Psi_m$-dependent probes was in fact due to differences in $\Delta\Psi_m$, we examined the possibility that different cristae within the same

mitochondrion could function as unique electrochemical domains and thus as independent bioenergetic units. We observed that, when imaging at high temporal resolution, laser-induced depolarization of elongated mitochondria did not tend to result in a simultaneous collapse of $\Delta\Psi_m$ across the entire organelle but rather in a rapid, wavelike depolarization. This discrepancy from earlier studies can be attributed to the fact that previous assays involving laser-induced damage did not image the mitochondria rapidly enough after

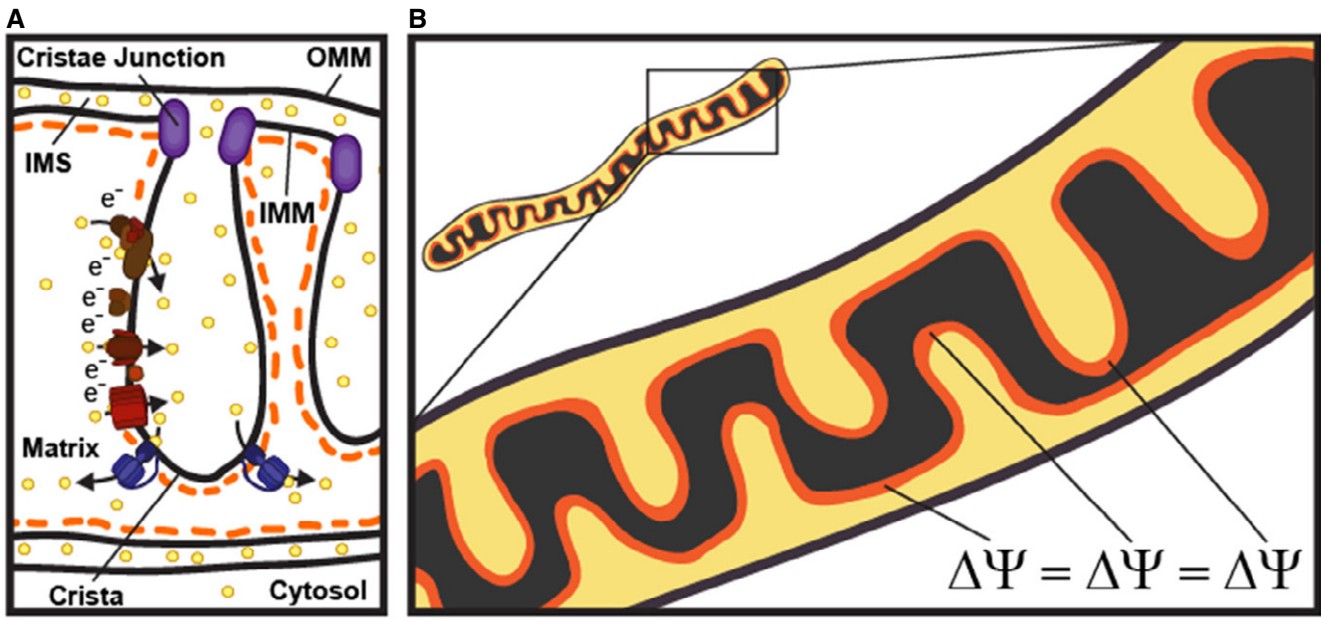

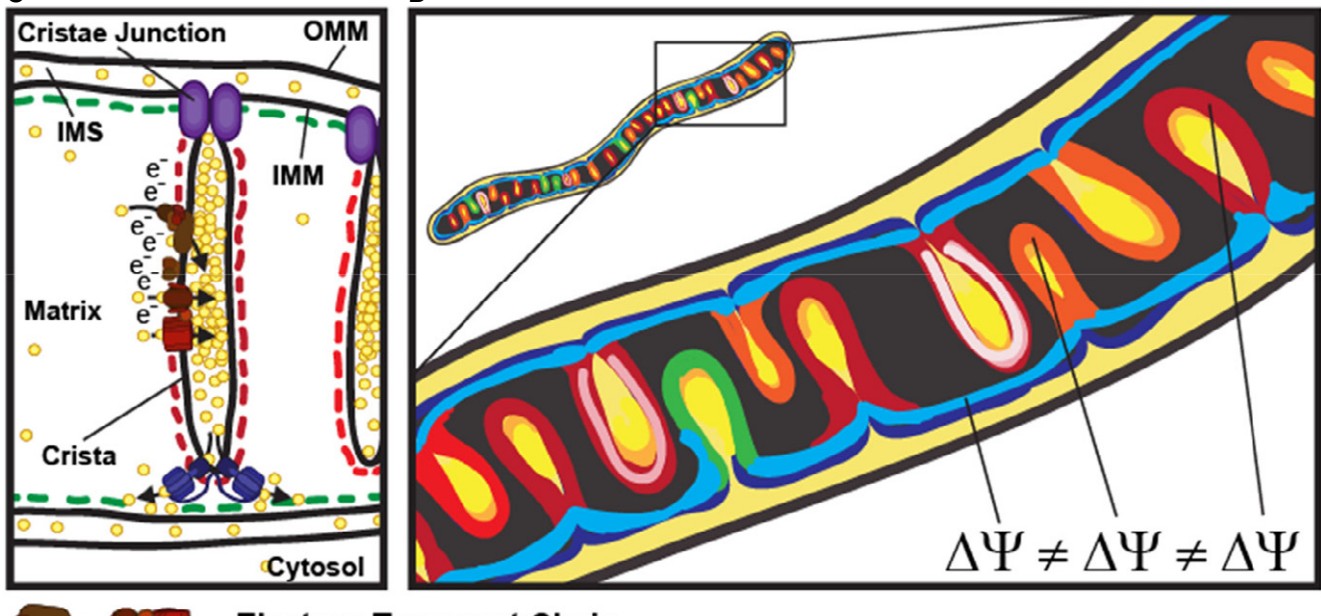

**Figure 11.**

**Figure 11. Model of cristae as individual bioenergetic units.**

A   An equipotential representation of the IMM. In this model, the IMM functions as a continuous electrical cable where cristae and IBM have similar $\Delta\Psi_m$ and where all cristae respond simultaneously to any change in $\Delta\Psi_m$ occurring in the mitochondrion where they reside. Transition from a hetero-potential to an equipotential, via cleavage of Opa1, for example, could play a role in normal mitochondrial physiology when high conductance and low ATP synthesis are favored, such as thermogenesis or adaptation to excess nutrient.

B   Cartoon showing IMM as equipotential.

C   Model of mitochondrial hetero-potential, showing cristae as individual bioenergetic units. Electrochemical autonomy of cristae protects the mitochondrion confronting a bioenergetic crisis, where dysfunction of one or more cristae would not inevitably cause the entire organelle to fail. In this model, cristae and IBM can have different membrane potentials. Closed CJs generate significant differences in charge between the cristae membranes and IBM, which could be related to the concentration of ETC subunits in the lateral side of the cristae and $F_1F_0$ ATP Synthase on the cristae rims.

D   Cartoon showing IMM as hetero-potential.

perturbing a small region of the organelle—for example, acquisition of a single frame expended 5–10 s (Amchenkova *et al*, 1988; Skulachev, 2001). Critical delays, either in the time it takes to acquire a single frame of a time series (Amchenkova *et al*, 1988) or in the duration of the perturbation itself (Glancy *et al*, 2015), will increase the likelihood of missing possible spatial differences in depolarization (originating close to the site of laser-induced damage). Our study suggests that, unless imaging at $\geq 1$ frame(s) per second, observing partial depolarization events is improbable. The tendency for one mitochondrion to exhibit localized depolarization indicates that a single organelle does not resemble an electrical wire (Fig 11A and B) but rather appears to function more like a configuration of interconnected batteries (Fig 11C and D).

Our high- or super-resolution imaging of the IMM stained with $\Delta\Psi_m$-dependent dyes is consistent with a model of cristae as individual, interconnected batteries. The $\Delta\Psi_m$ is not homogeneous along the IMM but shows significant differences between the cristae and IBM, demonstrating electrochemical discontinuity between the otherwise physically connected membranes. The relatively more negative $\Delta\Psi$ of the cristae membrane is expected to exert a stronger electrical force, keeping protons proximal to the cristae membrane. Indeed, our study can provide an explanation for heterogeneous distribution of protons along the cristae membrane and between the cristae and the IMS (Busch *et al*, 2013; Rieger *et al*, 2014; Pham *et al*, 2016). Following deletions of various proteins involved in regulating cristae architecture, including CJ formation, we observed a decrease in the difference between the $\Delta\Psi_m$ at the cristae membranes relative to the IBM. These data suggest that opening of CJs tends to equalize the $\Delta\Psi_m$ between cristae and IBM by a mechanism that requires further study. The generation of hyperpolarized vesicles, following different genetic perturbations that eliminate CJs, reinforces our general observation that increasing compartmentalization of the IMM, in turn, intensifies the electrochemical gradient. Lastly, our time-lapse imaging experiments show that the $\Delta\Psi_{Cr-IBM}$ of an individual crista is less variable over time compared to the $\Delta\Psi_{Cr-IBM}$ among different cristae within the same mitochondrion. Furthermore, we observe that, during transient depolarization, some cristae can maintain polarity despite the collapse in $\Delta\Psi_m$ of adjacent cristae. In summary, our data support a new paradigm where each crista maintains its own $\Delta\Psi_m$ that is different from both the IBM and from neighboring cristae.

This study raises interesting questions as to why mitochondria organize the $\Delta\Psi_m$ in this way. One advantage, for example, could be related to the fact that $\Delta\Psi_m$ constitutes the main energy available to drive protons through $F_1F_0$ ATP Synthase to produce ATP. As such, the localization of $F_1F_0$ ATP Synthase at the cristae rims appears to be advantageous in terms of proximity to the batteries. Another possible advantage could be compartmentalization of $\Delta\Psi_m$

in each crista may serve as a safeguard mechanism restricting the impact of localized damage. In the case of the equipotential model, where the inner membrane of the entire mitochondrion represents a single capacitor, a breach in membrane integrity in one crista would cause a collapse in voltage in all cristae and compromise the function of the whole organelle. If, on the other hand, the IMM could maintain numerous, discrete electrochemical gradients, like a group of batteries, then failure of one or more would not invariably jeopardize the entire mitochondrion. This may be of particular relevance in cells harboring a highly interconnected mitochondrial network as opposed to cells with less elongated and/or branched mitochondria. Furthermore, the hetero-potential model suggests that cristae with higher $\Delta\Psi_{Cr-IBM}$ could compensate for cristae with impaired function.

Hyperpolarized and depolarized IMM potentials are associated with different states of respiration. While both uncoupling and an increased rate of ATP synthesis dissipate $\Delta\Psi_m$, a decrease in ATP synthesis may result in hyperpolarization and increased ROS production. The hetero-potential model of the mitochondrion allows for different cristae to serve different functions. In this model, some cristae could be more dedicated to ATP synthesis, whereas neighboring cristae could play a role in ROS signaling. The hetero-potential model further allows for the consideration that different cristae may engage in primarily complex II vs. complex I respiration, which are associated with different membrane potentials and could be driven by different fuels.

In conclusion, our study identifies a new parameter of mitochondrial function: a mitochondrial hetero-potential arising from the compartmentalization of the $\Delta\Psi_m$ along the IMM. These findings have wide-ranging implications for human health, because many diseases and medical complications—for example, dominant optic atrophy (Amati-Bonneau *et al*, 2008; Zanna *et al*, 2008), ischemia-reperfusion injury (Birk *et al*, 2013), and even aging (Daum *et al*, 2013)—are associated with severe perturbations in cristae structure. Pathogenic changes to cristae architecture may undermine the autonomy of cristae bioenergetics, leading to the conversion of the IMM into an equipotential membrane. Consequently, damage inflicted to any isolated region may impact the bioenergetics of the entire mitochondrion, rendering the cell more vulnerable to mitochondrial toxicants and metabolic stress. Restoring the architecture of the IMM, therefore, could represent a viable approach to reestablishing the disparate electrochemical gradients that are the basis for normal mitochondrial function. This study emphasizes the need to directly explore the relationship between cristae structure and function to further elucidate the etiology of such diseases and, in turn, foster more effective therapies for treating them.

# Materials and Methods

### Cell culture

H1975 (sh-Scramble and sh-PTPMT1) cells were grown in RPMI-1640 (31800-022), supplemented with sodium bicarbonate, Pen/Strep, sodium pyruvate, HEPES, and 10% FBS, and cultured in 5% $CO_2$. Patient-derived fibroblasts were grown in DMEM (12100-046), supplemented with sodium bicarbonate, Pen/Strep, sodium pyruvate, HEPES, and 10% FBS, and cultured in 5% $CO_2$. L6, MEF, and HeLa cells were cultured in DMEM (12100-046), supplemented with sodium bicarbonate, Pen/Strep, sodium pyruvate, HEPES, and 10% FBS, and cultured in 5% $CO_2$. Hap1 cells were cultured in Iscove's modified Dulbecco's medium (IMDM) (I3390-500 ml), supplemented with 20% FBS, Pen/Strep, and 2 mM Glutamax. Primary cultures were isolated from unmodified C57BL6/J males and females, 12–28 weeks old, procured from Jackson laboratories and bred in our facility. Mice were provided with water and food *ad libitum*, and housed 2–5 mice per cage, 12-h light:dark cycle and at a room temperature of 22–24°C.

### KD and KO models

Human H1975 cells expressing shRNA targeting human PTPMT1 were generated using a pLKO.1 vector harboring shRNA (PTPMT1) Clone ID XM_374879.1-547s1c1 (Sigma-Aldrich). Lipofectamine 2000 (Thermo Fisher) with PLKO.1 shRNA (PTPMT1) plasmid, lentiviral packaging plasmids pMDLg/pRRE, pRSV-Rev, and enveloping plasmid pMD2.G (Addgene #12251, 12253, 12259) were used to transfect 293T cells to generate lentiviral particles. Forty-eight hours post-transfection, viral media was removed from 293T cells and centrifuged for 5 min at 300 *g*. Viral supernatant was then passed through 0.45-μM filter. Target cells were plated at 60% confluency and infected with 1× polybrene and viral supernatant. Viral media was removed and substituted for fresh culture media after 24 h, and incubated for an additional 24 h, followed by puromycin selection. Mic13 KO model was generated through Crispr/Cas9 genome editing (preprint: Kondadi *et al*, 2019), according to previously described methods (Anand *et al*, 2016). Mic10 and Mic60 KO cells were made by Horizon (UK) (preprint: Kondadi *et al*, 2019). Opa1 and Opa1/DRP1 were also generated through Crispr/Cas9 genome editing.

### Live-cell imaging

Cells were plated in CELLview 4-compartment glass-bottom tissue culture dishes (Greiner Bio-One, 627870), PS, 35/10 mm. Dyes (100 nM 10-*N*-nonyl acridine orange, 15 nM TMRE, 5 μM Rho123, and/or 200 nM MitoTracker Green; Invitrogen) were added to cell culture media and incubated 1–3 h prior to live-cell imaging with the alpha Plan-Apochromat 100×/1.46 Oil DIC M27 objective on the Zeiss LSM 880 with Airyscan. Prior to image analysis, raw .czi files were automatically processed into deconvoluted Airyscan images using the Zen software. HeLa cells imaged with STED were stained with 50 nM TMRM (Invitrogen) for 30 min before imaging. Leica SP8 LSM, fitted with STED module, was used to perform live-cell super-resolution imaging.

### Image analysis

Processed Airyscan images were analyzed using ImageJ (Fiji) software. Briefly, prior to cell cropping and quantification, background was subtracted from all images using a rolling ball filter = 50. After developing analysis protocols, we designed macros for high-throughput image quantification and analysis. For representative images in figures, we applied the Window/Level function when demonstrating relevant changes in pixel intensities; when specifically comparing cellular structures, we adjusted pixel intensities to optimally demonstrate relevant changes in structure. Images acquired with STED microscopy were deconvoluted using Huygens deconvolution software.

### Immunoblotting

SDS–PAGE assays were conducted by separating 20–30 μg of protein on precast polyacrylamide NuPAGE gels (Thermo Fisher) and transferred to nitrocellulose membranes in transfer buffer containing 10% methanol. Proteins were probed with the following primary and secondary antibodies: Opa1 1:1,000 (BD Biosciences, 612606); HSP60 1:10,000 (Abcam, ab46798); Drp1/DLP1 1:1,000 (BD Biosciences, 611112); PTPMT1 1:1,000 (Sigma-Aldrich, HPA043932); β-actin 1:10,000 (Cell Signaling Technology, BH10D10); TOM20 1:2,000 (Abcam, ab78547); Mic10/C1orf151 1:1,000 (MA5-26031); Mic13/C19orf70 1:1,000 (ProteinTech 25514-1-AP); Mic60/mitofilin 1:1,000 (ProteinTech 10179-1-AP); anti-mouse IgG, HRP-linked antibody (Cell Signaling Technology, 7076S); and anti-rabbit IgG, HRP-linked antibody (Cell Signaling Technology, 7074S). Proteins were detected and imaged using SuperSignal West Femto Maximum Sensitivity Substrate (Thermo Fisher) and ChemiDoc MP Imaging System (Bio-Rad), respectively. Densitometry was performed using ImageJ (Fiji).

### Statistical analysis

All statistical analyses were performed on GraphPad Prism and Microsoft Excel. Data sets were subjected to D'Agostino–Pearson omnibus and/or Shapiro–Wilk normality tests to assess whether data were normally distributed. Data were subjected to 2-tailed Student's *t*-tests, and *P* values < 0.05 were considered statistically significant. Error bars represent SEM, unless otherwise indicated. $N$ = the number of independent experiments. On average, ~ 20 cells were analyzed per independent experiment per condition. Statistical analysis was performed on the averages from independent experiments.

Expanded View for this article is available online.

### Acknowledgements

We would like to thank the referees and editor, who gave us outstanding advice and helped us build the concepts presented in this work. We thank Drs. Barbara Corkey, David Nicholls, Martin Picard, Gilad Twig, Gulcin Pekkurnaz, Ophry Pines, Victor Darley-Usmar, Fernando Abdulkader, György Hajnóczky, and Daniel Dagan for insightful discussions. We thank Samuel Itskanov for generating the Opa1 and Opa1&Drp1 DKO MEF cell lines and Dr. Yair Anikster for providing human fibroblasts. We also thank Drs. Mingqi Han and Alejandro Martorell Riera for help with reagents and cell lines. We thank the Centre for

Advanced Imaging (CAi) at HHU, Düsseldorf, for providing facilities for STED imaging. ASR was supported by the Deutsche Forschungsgemeinschaft (DFG) grant RE 1575/2-1 and SFB 974 Project B09; AKK and RA/ASR were supported by the Medical faculty, Heinrich Heine University Düsseldorf FoKo-37/2015 and Foko-02/2015, respectively. OSS is funded by NIH-NIDDK 5-RO1DK099618-02. ML and MS are funded by UCLA Department of Medicine Chair commitment and UCSD/UCLA Diabetes Research Center grant, NIH P30 DK063491. DBS is funded by an NIH/NCI R01 CA208642-01. STB was supported by an NIH T32 training grant HL072752.

## Author contributions

DMW and MS designed the study, made the figures, and wrote the article. DMW and MS performed all of the Airyscan imaging, Western blotting, as well as data analysis. STB generated the sh-Scramble and sh-PTPMT1 cell lines. AKK performed the STED imaging. RA created the Mic13 KO cell line. ASR, AMB, and DBS provided valuable recommendations, contributing to the study design and reagents. OSS and ML provided supervision, designed the study, and wrote the article.

## Conflict of interest

The authors declare that they have no conflict of interest.

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
