## [Review Process File · The EMBO Journal]

Individual mitochondrial cristae function as independent bioenergetic units.

Dane M. Wolf, Mayuko Segawa, Arun Kumar Kondadi, Ruchika Anand, Sean T. Bailey, Andreas S. Reichert, Alexander M. van der Blik, David B. Shackelford, Marc Liesa, Orian S. Shirihai.

Review timeline:

Submission date:	1 st November 2018
Editorial Decision:	17 th December 2018
Revision received:	10 th June 2019
Editorial Decision:	9 th July 2019
Revision received:	12 th August 2019
Accepted:	5 th September 2019

Editor: Elisabetta Argenzio

Transaction Report:

1st Editorial Decision

17th December 2018

Thank you for submitting your manuscript on a role for mitochondrial cristae in promoting the electrochemical gradient that drives ATP production to the EMBO Journal. My apologies for the extended duration of the review process for this manuscript. Three referees were originally assigned to your manuscript but one of them did not return his/her report even after repeated messages. In light of the two reports available, I am afraid we have concluded that we cannot offer publication here.

As you will see, while the referees find the study potentially interesting and recognize the quality of your data, they also raise a number of major issues that preclude publication in The EMBO Journal. In particular, they both point to a marked discrepancy between the title/abstract and the content of the manuscript and thus find the main conclusions emphasized in the abstract not sufficiently supported by the data. In addition, they feel that the choice of cardiolipin knock down as model system is too drastic leaving the resulting phenotypes predictable.

Given these opinions from trusted experts in the field and the extensive additional data that would be needed to substantiate your model, I am afraid that we cannot offer to invite a revised version of your manuscript at the EMBO Journal.

However, considering the potential interest of your findings, I would be willing to reconsider your study as a new submission at a later time if the referees' concerns could be fully addressed and their suggestions implemented. Please note that the novelty of your study will be taken into consideration at the time of resubmission.

REFeree REPORTS

Referee #1:

This is an interesting and original piece of work. The experiments have been performed with care and include data obtained with sophisticated novel approaches, yielding original results that deserve publication, yet I have concerns that in my opinion make the manuscript unsuitable for EMBO Journal in its present form.

Reading the title and abstract I expected quite a different manuscript. The title refers exclusively, and the abstract almost exclusively, to heterogeneity of cristae structure and membrane potential in individual wild-type mitochondria, yet most of the manuscript is in fact about the features of mitochondria and cells from their PTPMT1 knockdown counterpart (Figures 1 to 5, part of Figure 7 and all of the Supplementary Figures). In the context of heterogeneity, the PTPMT1 KD cells (which do not display the subcompartmentation) are used as a proof of concept "in reverse", i.e. as the demonstration that the original, wild-type phenotype can be lost and that it therefore reflects the true properties of mitochondria in situ. Yet the choice of the KD model appears quite drastic, in that cardiolipin is essential for proper function of so many mitochondrial proteins that the resulting phenotype is, somewhat predictably, very severe. The detailed characterization of the consequences of PTPMT1 KD on Opa1 and Mfn1 expression, fragmentation by Alexidine and the profound inhibition of respiration in the KD cells are interesting but not directly related to the issue of membrane potential heterogeneity in normal mitochondria.

The data on wild-type cells appear in contrast with those reported in Nature by Glancy et al. about the presence of an electrically connected mitochondrial reticulum in skeletal muscle cells [1]. I am surprised that this article is not discussed, because the present data and those reported there suggest that we may be dealing with extremes (full connectivity of inner membranes versus electrically isolated crista sections) that may either depend on constitutive differences (muscle versus other cells) or on subcompartment regulation by proteins such as those described here to be indeed altered in PTPMT1 KD, such as Opa1 or mitofusins, or perhaps other crista junction proteins of the MICOS complex. I think that an effort should be made to assess this important problem by using the technique in other cell types, particularly skeletal muscle. While certainly interesting, the current findings cannot be described as a new mitochondrial paradigm at the present stage. While of course decisions about revision rest with the Authors, my suggestion would be to carry out the suggested experiments in muscle cells and/or other cell types, and write a perhaps shorter manuscript focusing on the resulting findings and their functional implications.

Irrespective of the final revision, the abstract (and other parts of the text) needs refocusing, as the compartmentalization cannot contribute to "create" the proton gradient. Formation of the proton gradient depends on proton pumping and on an insulating inner membrane, not on the shape of the cristae. Rather than pointing at specific issues I have modified the original abstract to make my point - of course only suggestions.

"Invaginations of the inner mitochondrial membrane (IMM), called cristae, may promote the formation of compartments experiencing different mitochondrial membrane potential ($\Delta\Psi_m$) and ΔpH values, which could in turn impact on ion and metabolite transport in specific regions of the organelles. Such a structural arrangement and its potential impact on the submitochondrial distribution of the proton electrochemical gradient, which drives the rotor of FOF1 ATP synthase producing ATP, has never been tested directly because of the inability to resolve individual cristae in mitochondria in situ. To address this question, we loaded mitochondria with the $\Delta\Psi_m$ -responsive dye, TMRE, and performed Airyscan imaging, a new approach enabling resolution of cristae in live cells. We made the remarkable observation that the $\Delta\Psi_m$ is significantly higher at the cristae (than, something is missing here). Knocking down of PTPMT1, a key enzyme in the cardiolipin biosynthesis pathway, disrupted cristae structure, blunted these local peaks in $\Delta\Psi_m$ and induced haphazard movements of the IMM. Altogether, our data reveal the existence of functionally discrete IMM cristae domains where the electrical and chemical gradient of protons can be independently modulated."

Specific points

Page 3, last line of 1st paragraph "many questions remain as to how the structure of the IMM contributes to the $\Delta\psi$." should be reworded as suggested above for the abstract. Next line, why "theoretical"? I think that this is based on experiments rather than theories; last line, not "through the pore" but "through the proton channel of the c ring", which is not a pore.

Page 4, first sentence isn't clear and the energy is from $\Delta\psi$ not $\Delta\psi_{\text{psi}}$ only.

Page 11, I am not sure that the use of the term "fermentation" is appropriate in this context.

Page 13, 1st line should be "bind" rather than "bond"; 3rd line Les Loew should be acknowledged for his pioneering work on the use of TMRE imaging to quantitatively assess the mitochondrial membrane potential in situ by imaging techniques [2,3].

Page 15, 1st line "electrically dependent" should be "dependent on local differences in mitochondrial membrane potential", also 6 lines below better "dependent on differences in magnitude of the electrical gradient"; line 8 remove "Indeed".

Page 16, line 6, again credit should be given (also) to Loew et al [3].

References

1. Glancy B, Hartnell LM, Malide D, Yu ZX, Combs CA, Connelly PS, Subramaniam S & Balaban RS (2015) Mitochondrial reticulum for cellular energy distribution in muscle. *Nature* 523, 617-620.
2. Farkas DL, Wei MD, Febroriello P, Carson JH & Loew LM (1989) Simultaneous imaging of cell and mitochondrial membrane potentials. *Biophys J* 56, 1053-1069.
3. Loew LM, Tuft RA, Carrington W & Fay FS (1993) Imaging in five dimensions: time-dependent membrane potentials in individual mitochondria. *Biophys J* 65, 2396-2407.

Referee #3:

In this work, the authors induced knockdown (KD) or inhibition of PTPMT1, an enzyme of the cardiolipin pathway, in cell cultures. Cardiolipin is a phospholipid specific to mitochondria.

They showed that these interventions caused loss of respiratory function, loss of mitochondrial membrane potential, altered expression of OXPHOS proteins, change in mitochondrial fusion activity, and changes in the expression and processing of proteins involved in the fusion process. The quality of the data is high but most of these observations were more or less expected given the likely loss of cardiolipin.

In addition, the authors made observations that suggested to them heterogeneity in the intramitochondrial distribution of the electrochemical potential. However, there is a concerning disconnect between the abstract/introduction and the main body of the paper. While the abstract emphasizes the intramitochondrial distribution of the membrane potential, most of the paper is focused on various other effects of PTPMT1-KD. The main conclusion emphasized in the abstract is not sufficiently supported by the data in my opinion.

MAJOR POINTS

1. To me the central conclusion of the paper that "the $\Delta\Psi_m$ is significantly higher at the cristae" seems trivial. The electrochemical potential arises obviously from a difference in proton concentration across the inner mitochondrial membrane. Since the cristae contain most of the inner membrane, it is not surprising that they also are the site of the electrochemical potential. The question would be whether other segments of the inner membrane, such as the boundary inner membrane or the cristae junctions, contribute to the electrochemical potential but this question is not addressed in the paper. The observation that TMRE fluorescence co-localized with NAO

fluorescence but not with mtPAGFP fluorescence, only shows that TMRE binds to the membrane but does not dissolve in the matrix, which again is not surprising. From my reading of the manuscript, I do not see any evidence for lateral heterogeneity in the electrochemical potential across the surface of the inner mitochondrial membrane.

2. The application of the new Airyscan technology has probably underutilized potential but I'm not sure that the increase in resolution is sufficient to actually quantify the density of cristae. Electron micrographs show cristae often within 10-100 nm of each other, whereas the images in the manuscript show cristae at a width of 100-200 nm at best. I'm not an expert in fluorescence microscopy or the associated data processing, but would like to voice this concern.

3. NAO is a poor choice for comparing the cristae density between PTPMT1-KD and control (Fig. 1) because the former does not contain cardiolipin, which is the main target of NAO. The authors point out correctly that NAO also binds to other (negatively charged) lipids but losing CL, the lipid that NAO has the highest affinity for, is a set-up for underestimating the cristae in the PTPMT1-KD.

4. Since the thrust of the paper is on the characterization of the PTPMT1-KD, it is necessary to quantify the proximal effect of PTPMT-KD, i.e. the changes in mitochondrial lipid composition. This would tell us about the extent of the presumed decrease in cardiolipin and about the lipid(s) cardiolipin is replaced by.

MINOR POINTS

- What is the reason for using "patient-derived fibroblasts and parental H1975 cells"?
- It has been hypothesized that CL is concentrated at the tips of the cristae but experimental evidence is lacking (introduction).

Major concerns Referee #1:

- 1) This is an interesting and original piece of work. The experiments have been performed with care and include data obtained with sophisticated novel approaches, yielding original results that deserve publication, yet I have concerns that in my opinion make the manuscript unsuitable for EMBO Journal in its present form.**

We thank the reviewer for highlighting our careful experimental performance and the sophistication of our novel approaches. We have addressed all the concerns raised by this reviewer and we are grateful for his/her remarks, which led to a substantial improvement in our manuscript.

- 2) Reading the title and abstract I expected quite a different manuscript. The title refers exclusively, and the abstract almost exclusively, to heterogeneity of cristae structure and membrane potential in individual wild-type mitochondria, yet most of the manuscript is in fact about the features of mitochondria and cells from their PTPMT1 knockdown counterpart (Figures 1 to 5, part of Figure 7 and all of the Supplementary Figures). In the context of heterogeneity, the PTPMT1 KD cells (which do not display the subcompartmentation) are used as a proof of concept "in reverse", i.e. as the demonstration that the original, wild-type phenotype can be lost and that it therefore reflects the true properties of mitochondria in situ. Yet the choice of the KD model appears quite drastic, in that cardiolipin is essential for proper function of so many mitochondrial proteins that the resulting phenotype is, somewhat predictably, very severe. The detailed characterization of the consequences of PTPMT1 KD on Opa1 and Mfn1 expression, fragmentation by Alexidine and the profound inhibition of respiration in the KD cells are interesting but not directly related to the issue of membrane potential heterogeneity in normal mitochondria.***

We agree with the reviewer's remarks that the previous organization and writing style resulted in the manuscript having two related narratives: 1) The demonstration of the heterogeneity of membrane potential in the individual cristae of a single mitochondrion; and 2) The effects of PTPMT1 KD on cristae structure, mitochondrial dynamics, and function. In consultation with the editor, we have revised the manuscript to focus on the first narrative. In this revised version, we have more than doubled the amount of data to expand the first narrative, including novel models of cristae-junction manipulations, as they support the existence of membrane potential heterogeneity. Furthermore, we now provide new evidence in different cell lines and use different mitochondria-labelling strategies, which demonstrate:

- a) A difference in membrane potential ($\Delta\Psi_m$) exists between the inner boundary membrane (IBM) and cristae in various cell types (Fig. 2). We have successfully quantified this difference in mV in various cell types, including primary hepatocytes (Fig. 3); and, we show that these differences in $\Delta\Psi_m$ heterogeneity can be increased by oligomycin and decreased by FCCP (Fig. 4B,C).
- b) Each crista is an independent bioenergetic unit. The evidence supporting this conclusion was obtained using different genetic manipulations (including Mic13-, Mic60-, Mic10- and OPA1-KO models) (Fig. 5), resulting in the alteration of cristae junctions and structure and decreased heterogeneity $\Delta\Psi_m$ of cristae relative to IBM. We also used laser-induced depolarization (Fig. 6A-F) and time-lapse imaging approaches (Fig. 6G-L) to demonstrate that different cristae can maintain different $\Delta\Psi_m$, supporting a new paradigm of the $\Delta\Psi_m$ (Fig. 7C,D).

3) *The data on wild-type cells appear in contrast with those reported in Nature by Glancy et al. about the presence of an electrically connected mitochondrial reticulum in skeletal muscle cells [1]. I am surprised that this article is not discussed, because the present data and those reported there suggest that we may be dealing with extremes (full connectivity of inner membranes versus electrically isolated crista sections) that may either depend on constitutive differences (muscle versus other cells) or on subcompartment regulation by proteins such as those described here to be indeed altered in PTPMT1 KD, such as Opa1 or mitofusins, or perhaps other crista junction proteins of the MICOS complex. I think that an effort should be made to assess this important problem by using the technique in other cell types, particularly skeletal muscle. While certainly interesting, the current findings cannot be described as a new mitochondrial paradigm at the present stage. While of course decisions about revision rest with the Authors, my suggestion would be to carry out the suggested experiments in muscle cells and/or other cell types, and write a perhaps shorter manuscript focusing on the resulting findings and their functional implications.*

As suggested by the reviewer, we have expanded our efforts and analyzed MEFs, HeLa, Hap1, L6 myoblasts, primary mouse hepatocytes, and human fibroblasts, as well as Opa1-, Mic13-, Mic10- and Mic60-KO cells (MICOS complex) (Figs. 2, 3, 5, & 6). We are also thankful to the reviewer for raising the absence of discussion of the Glancy and Skulachev studies showing the electrically connected mitochondrial reticulum. We have performed new experiments that reconcile our findings with previous data by Glancy and Skulachev (now references are included in the Introduction, pages 3 and 4; in the Results, pages 8, 14, 15, and 16; and in the Discussion, page 20). By increasing the frequency of image acquisition after a laser-induced depolarization event in a long mitochondrion (every ~0.15 sec/frame compared to ~5-10 sec/frame, in these previous studies), we observe that depolarization occurs in a wavelike (i.e., gradual) fashion,

starting from the site of laser exposure (Fig. 6A-F). In other words, an instantaneous collapse in voltage due to the mitochondrion functioning as an uninterrupted electrical conduit is incompatible with a wavelike depolarization, indicating that there are discrete domains of the electrochemical gradient along the IMM.

- 4) Irrespective of the final revision, the abstract (and other parts of the text) needs refocusing, as the compartmentalization cannot contribute to "create" the proton gradient. Formation of the proton gradient depends on proton pumping and on an insulating inner membrane, not on the shape of the cristae. Rather than pointing at specific issues I have modified the original abstract to make my point - of course only suggestions.**

"Invaginations of the inner mitochondrial membrane (IMM), called cristae, may promote the formation of compartments experiencing different mitochondrial membrane potential ($\Delta\Psi_m$) and ΔpH values, which could in turn impact on ion and metabolite transport in specific regions of the organelles. Such a structural arrangement and its potential impact on the submitochondrial distribution of the proton electrochemical gradient, which drives the rotor of FOF1 ATP synthase producing ATP, has never been tested directly because of the inability to resolve individual cristae in mitochondria in situ. To address this question, we loaded mitochondria with the $\Delta\Psi_m$ -responsive dye, TMRE, and performed Airyscan imaging, a new approach enabling resolution of cristae in live cells. We made the remarkable observation that the $\Delta\Psi_m$ is significantly higher at the cristae (than, something is missing here). Knocking down of PTPMT1, a key enzyme in the cardiolipin biosynthesis pathway, disrupted cristae structure, blunted these local peaks in $\Delta\Psi_m$ and induced haphazard movements of the IMM. Altogether, our data reveal the existence of functionally discrete IMM cristae domains where the electrical and chemical gradient of protons can be independently modulated."

We thank the reviewer for the excellent suggestions and comments on the abstract. We have carefully followed these recommendations. As we more than doubled the number of data panels demonstrating the heterogeneity and independence of individual cristae bioenergetics in a single mitochondrion, we ended up completely rewriting the abstract to highlight these new findings (see below). In this regard, we also completely agree with the reviewer's point that compartmentalization cannot by itself create the proton gradient. Indeed, our new experiments demonstrate the independency of cristae bioenergetics and $\Delta\Psi_m$ differences between the IBM and the cristae. We do not draw any conclusions on the effect of cristae structure on proton pumping and proton gradients. We thus have acknowledged this limitation in the discussion and carefully revised our wording to remove any text suggesting that compartmentalization/cristae structure by itself contributes to create the $\Delta\Psi_m$. Our new abstract reads as:

“The inner mitochondrial membrane (IMM) is considered to function as an electrical cable, where neighboring cristae and inner boundary membrane (IBM) carry the same membrane potential ($\Delta\Psi_m$). The sequestration of OXPHOS components in cristae membranes, however, necessitates a re-examination of the equipotential representation of the IMM. Here, we developed an approach to monitor $\Delta\Psi_m$ at the resolution of individual cristae. We found that the IMM was divided into segments with distinct $\Delta\Psi_m$, corresponding to cristae and IBM. $\Delta\Psi_m$ was higher at the cristae compared to IBM. Treatment with oligomycin increased, whereas FCCP decreased, $\Delta\Psi_m$ heterogeneity along the IMM. Moreover, impairment of cristae structure through deletion of MICOS-complex components or Opa1 diminished this intramitochondrial heterogeneity of $\Delta\Psi_m$. Lastly, we determined that different cristae within the individual mitochondrion can have discrete membrane potentials and that interventions causing acute depolarization may affect some cristae while sparing others. Altogether, our data support a new model in which cristae within the same mitochondrion behave as independent bioenergetic units, preventing the failure of specific cristae from spreading dysfunction to the rest.”

Minor concerns Referee #1:

- 5) Page 3, last line of 1st paragraph "many questions remain as to how the structure of the IMM contributes to the Δp ." should be reworded as suggested above for the abstract. Next line, why "theoretical"? I think that this is based on experiments rather than theories; last line, not "through the pore" but "through the proton channel of the c ring", which is not a pore.**

The introduction has been rewritten, and these statements have been removed.

- 6) Page 4, first sentence isn't clear and the energy is from Δp not $\Delta\Psi$ only.**

We have included this sentence in page 3 of the Introduction to clarify this: “The high concentration of hydrogen ions in the intermembrane space (IMS) constitutes the proton motive force (Δp), made up of electrical ($\Delta\Psi_m$) and chemical (ΔpH) potential energy.”

- 7) Page 11, I am not sure that the use of the term "fermentation" is appropriate in this context.**

This word has been removed.

- 8) Page 13, 1st line should be "bind" rather than "bond"; 3rd line Les Loew should be acknowledged for his pioneering work on the use of TMRE imaging to quantitatively assess the mitochondrial membrane potential in situ by imaging techniques [2,3].**

This has been corrected and Loew references have been included in the Results, pages 8, 10, 12; and in the Discussion, page 19.

- 9) ***Page 15, 1st line "electrically dependent" should be "dependent on local differences in mitochondrial membrane potential", also 6 lines below better "dependent on differences in magnitude of the electrical gradient"; line 8 remove "Indeed".***

This text has been removed and this section substantially rewritten.

- 10) ***Page 16, line 6, again credit should be given (also) to Loew et al [3].***

Credit has been given now in the revised section of the discussion, now in page 19.

References

1. Glancy B, Hartnell LM, Malide D, Yu ZX, Combs CA, Connelly PS, Subramaniam S & Balaban RS (2015) Mitochondrial reticulum for cellular energy distribution in muscle. *Nature* 523, 617-620.
2. Farkas DL, Wei MD, Febroriello P, Carson JH & Loew LM (1989) Simultaneous imaging of cell and mitochondrial membrane potentials. *Biophys J* 56, 1053-1069.
3. Loew LM, Tuft RA, Carrington W & Fay FS (1993) Imaging in five dimensions: time-dependent membrane potentials in individual mitochondria. *Biophys J* 65, 2396-2407.

Major concerns Referee #3:

- 1) ***In this work, the authors induced knockdown (KD) or inhibition of PTPMT1, an enzyme of the cardiolipin pathway, in cell cultures. Cardiolipin is a phospholipid specific to mitochondria. They showed that these interventions caused loss of respiratory function, loss of mitochondrial membrane potential, altered expression of OXPHOS proteins, change in mitochondrial fusion activity, and changes in the expression and processing of proteins involved in the fusion process. The quality of the data is high but most of these observations were more or less expected given the likely loss of cardiolipin.***

We thank the reviewer for highlighting the quality of our data. We also greatly appreciate this reviewer's remarks, because we think that addressing his/her points has helped us to focus our narrative and substantially improve our manuscript.

We have addressed the limitation raised by the reviewer relating to the expected outcome resulting from PTPMT1 inhibition by performing new experiments, demonstrating that: 1) $\Delta\Psi_m$ is higher at the cristae, when compared to the inner boundary membrane (IBM) (Figs. 2 & 3). 2) The difference in $\Delta\Psi_m$ between the cristae and the IBM is significantly decreased by deleting 3 different components of the MICOS complex, separately, as well as OPA1 (Fig. 5). These results indicate that the difference in $\Delta\Psi_m$ is decreased by other genetic manipulations altering cristae structure that are less severe than PTPMT1 KD. 3) We demonstrated for the first time that individual cristae within a single mitochondrion are independent bioenergetic units, which include time-lapse imaging as well as laser-induced depolarization experiments at rapid acquisition rates (~0.15 sec/frame) (Fig. 6).

2) In addition, the authors made observations that suggested to them heterogeneity in the intramitochondrial distribution of the electrochemical potential. However, there is a concerning disconnect between the abstract/introduction and the main body of the paper. While the abstract emphasizes the intramitochondrial distribution of the membrane potential, most of the paper is focused on various other effects of PTPMT1-KD. The main conclusion emphasized in the abstract is not sufficiently supported by the data in my opinion.

We thank the reviewer for this comment. This concern regarding two narratives was also shared by reviewer 1 and by the editor. In consultation with the editor, we have focused the manuscript on the intramitochondrial distribution of the membrane potential. In the revised manuscript, we have a large set of new data to substantiate that: 1) The $\Delta\Psi_m$ at cristae is significantly higher than at IBM in various cell lines (Figs. 2 & 3). 2) Oligomycin increases, whereas FCCP decreases, the $\Delta\Psi_m$ at cristae relative to IBM (Fig. 4B-D). 3) Knockout of components of the MICOS complex as well as OPA1 decrease the difference in $\Delta\Psi_m$ between cristae and IBM, demonstrating that proteins regulating cristae structure and junctions are required to maintain this heterogeneity (Fig. 5). 4) Laser-induced depolarizations (Fig. 6A-F) as well as random flickering events (Fig. 6I-L) show that specific cristae along a single mitochondrion can remain polarized while other cristae depolarize; we also show that individual cristae have less variability in $\Delta\Psi_m$ over time compared to variability in $\Delta\Psi_m$ among different cristae (Fig. 6G,H), altogether demonstrating the existence of bioenergetic independency among cristae subunits.

3) To me the central conclusion of the paper that "the $\Delta\Psi_m$ is significantly higher at the cristae" seems trivial. The electrochemical potential arises obviously from a difference in proton concentration across the inner mitochondrial membrane. Since the cristae contain most of the inner membrane, it is not surprising that they also are the site of the electrochemical potential. The question would be whether other segments

of the inner membrane, such as the boundary inner membrane or the cristae junctions, contribute to the electrochemical potential but this question is not addressed in the paper. The observation that TMRE fluorescence co-localized with NAO fluorescence but not with mtPAGFP fluorescence, only shows that TMRE binds to the membrane but does not dissolve in the matrix, which again is not surprising. From my reading of the manuscript, I do not see any evidence for lateral heterogeneity in the electrochemical potential across the surface of the inner mitochondrial membrane.

We appreciate this excellent comment. We have measured the $\Delta\Psi_m$ at the cristae and IBM and detect a significant difference, with $\Delta\Psi_m$ at the cristae being higher than IBM (Fig 3). Furthermore, we find that deleting Opa1 or components of the MICOS complex (i.e., Mic10, Mic60 and Mic13) decrease the difference in $\Delta\Psi_m$ between the cristae and the IBM (Fig. 5). Thus, we now provide evidence of heterogeneity across the surface of the inner mitochondrial membrane that is diminished as a result of different genetic manipulations disrupting the cristae junction and structure.

- 4) ***The application of the new Airyscan technology has probably underutilized potential but I'm not sure that the increase in resolution is sufficient to actually quantify the density of cristae. Electron micrographs show cristae often within 10-100 nm of each other, whereas the images in the manuscript show cristae at a width of 100-200 nm at best. I'm not an expert in fluorescence microscopy or the associated data processing, but would like to voice this concern.***

We have addressed this concern by substantiating our evidence with new data acquired with STED microscopy (Fig. 3G-I); and, by emphasizing the scale bars in the Airyscan images, we show that we can resolve cristae ≤ 100 nm from each other (Fig. 1B). In addition, it is possible that the fixation and dehydration procedures required to perform some EMs could change the distances between membranes. Thus, it is possible that distances observed in living cells are not maintained after cell fixation and dehydration.

- 5) ***NAO is a poor choice for comparing the cristae density between PTPMT1-KD and control (Fig. 1) because the former does not contain cardiolipin, which is the main target of NAO. The authors point out correctly that NAO also binds to other (negatively charged) lipids but losing CL, the lipid that NAO has the highest affinity for, is a set-up for underestimating the cristae in the PTPMT1-KD.***

We appreciate this advice, and we removed the quantification of cristae density. We have addressed this concern by using 4 new genetic manipulations, including KO of 3 components of the MICOS complex as well as OPA1, which do not target cardiolipin

synthesis (Figs 1 & 5). These data demonstrate that cristae alterations can be detected independently of the CL depletion associated with PTPMT1 KD.

- 6) *Since the thrust of the paper is on the characterization of the PTPMT1-KD, it is necessary to quantify the proximal effect of PTPMT-KD, i.e. the changes in mitochondrial lipid composition. This would tell us about the extent of the presumed decrease in cardiolipin and about the lipid(s) cardiolipin is replaced by.***

This would be an excellent experiment to understand the effects of PTPMT1 KD. We have refocused our manuscript, however, on the question of heterogeneity of membrane potential between cristae and IBM. In the current manuscript, we limit the use PTPMT1 KD cells as one of several examples of changes in cristae structure (Fig 1). To more directly examine the relationship between cristae junction and membrane potential heterogeneity, we focused on components of the MICOS complex as well as Opa1. Please note that Fig. 5 shows that IMM architecture is altered in cells lacking, separately, Mic13, Mic60, and Mic10, as well as Opa1.

Minor points Referee #3:

- 7) *What is the reason for using "patient-derived fibroblasts and parental H1975 cells"?***

In our previous manuscript, we were using these cell lines to test whether treatment with the inhibitor of PTPMT1 (i.e., Alexidine) would produce the same effects in various cell lines. In our current manuscript, however, we largely removed the data relating to PTPMT1, including the Alexidine treatments.

We are now using human fibroblasts to try to repeat the kind of experiments run by Skulachev ~30 years ago, when the cable theory was first tested. Their elongated mitochondrial morphology presents an optimal system in which to perform laser-induced depolarization experiments (Fig. 6C-F).

- 8) *It has been hypothesized that CL is concentrated at the tips of the cristae but experimental evidence is lacking (introduction).***

We thank the reviewer for bringing this to our attention. We agree that experimental evidence for this statement is lacking. Given that our manuscript is not focused on the role of CL in cristae heterogeneity anymore, we deleted this section from the Introduction.

Thank you for submitting a revised version of your manuscript, which has now been seen by the original referees.

As you will see in the reports shown below, both reviewers find that the manuscript has strongly improved after revision, but they nevertheless request you to expand some points in the discussion and to correct misconceptions throughout the text.

In addition to solving the above-mentioned referees' points, there are a few editorial issues about the text and the figures that I need you to address before we can officially accept the manuscript.

REFEREE REPORTS

Referee #1:

The manuscript has been thoroughly revised and my main general concerns have been effectively addressed experimentally. I would like to commend the authors for this excellent job. However, I still have a major concern (not requiring experiments) that I believe needs to be addressed.

Beginning of the Introduction, the statement "The high concentration of hydrogen ions in the intermembrane space (IMS) constitutes the proton motive force (Δp), made up of electrical ($\Delta\Psi_m$) and chemical (ΔpH) potential energy (Mitchell, 1961)" is incorrect. Formation and maintenance of the mitochondrial proton gradient depends on both proton pumping and on the insulating properties of the inner membrane. The protonmotive force does not depend on the extramitochondrial (intermembrane space) concentration of protons. This is a misconception that in the early days was used as an argument against chemiosmosis. The comment was that proton pumping into the cytosol was like proton pumping into the ocean, implying that a proton gradient could not build up because of proton diffusion! The (still valid) point of Mitchell was that the proton gradient would form even if extramitochondrial protons were diluted into the ocean, because the extruded proton would still "leave behind" a more electronegative matrix (i.e., a membrane potential DIFFERENCE would still be formed because there is little or no permeability to anions and other charged species). The chemical (ΔpH) and electrical ($\Delta \psi$) components actually balance in opposite directions, and the membrane potential difference decreases when the ΔpH increases (i.e., when a measurable acidification takes place like in the compartments described here).

A few lines below the authors state: "To date, the prevailing view of the Δp is that, after respiratory complexes I, III, and IV pump protons from the matrix across the IMM, the protons are free to move throughout the IMS (Amchenkova, 1988; Mannella et al., 2013; Skulachev, 2001). According to this model, random distribution of protons throughout the IMS would lead to an equal distribution of positive charges all along the outward-facing leaflet of the IMM. Consequently, the difference in proton concentration and, in turn, the difference in charge, between the IMS and matrix would be the same at any point along the IMM. This model provides the theoretical basis that the $\Delta\Psi_m$ constitutes an equipotential - i.e., each mitochondrion, at any given time, represents a uniform voltage." Again, this is not the point. The outer membrane is very permeable to protons, and therefore IMS protons equilibrate rapidly (in the absence of constraints; in isolated mitochondria pumped protons are readily measured in solution) and the ΔpH is small (also due to buffering of matrix protons by phosphate and organic anions). In other words, the "equipotential" membrane would not rely on proton diffusion, but rather on the ultrastructure of mitochondria and on the homogeneity of the membrane properties (once the proton is extruded, the voltage "travels" along the membrane and is not "carried" by protons).

Also the discussion suffers from what appears to be a misconception. For example, page 22 "This study raises interesting questions as to why mitochondria organize the Δp in this way. One possibility is that the OMM is relatively porous compared to the IMM. If protons pumped across IMM by the respiratory complexes were permitted to diffuse throughout the IMS, it is likely that they would readily escape into the cytosol. Confining protons in the cristae lumen by increasing

resistance at the CJ would, on the other hand, concentrate protons near the array of FoF1 ATP Synthase dimers, increasing the probability of proton translocation back into the matrix and, in turn, promoting ATP production." ATP production is driven as easily by an electrical potential difference, which drags back the protons irrespective of the delta pH, so this is not a valid argument. What drives ATP synthesis is the proton electrochemical GRADIENT, not only the delta pH.

Also at page 22, "Another possibility that the IMM has evolved different electrical domains could be to regulate import and export of proteins, metabolites, ions, etc. An equipotential could interfere with the movement of various essential molecules in and out of the organelle: if there were an equally high proton concentration at any point throughout the IMS, then the relatively high concentration of positive charges could act as an electrical barrier against other positively charged molecules or ions needing to cross the membrane. A hetero-potential shows decreased proton concentration near the IBM, which could decrease resistance to transport of various materials in and out of the organelle." These speculations are excessive, and I don't think that they are needed. It is fine to say that physical constraints between cristae may lead to individual "units" (as measured here), but the acidification of these compartments (not measured here) would be the consequence, not the cause of the heterogeneity. And it should be noted that the higher the delta pH (i.e. the local acidification) the lower the delta psi and vice versa. I think that the "barriers" at the crista junctions should rather be viewed as electrical barriers, similar to myelinated regions of the neuron, i.e. structures that prevent lateral propagation of the electrical signal to the next structure. This may well be (and probably is) a dynamic process. Certainly heterogeneity opens the door to subtle changes in microdomains that may then fulfill different local requirements. What is the meaning is hard to guess at present, but I feel that speculations should be kept to a minimum.

Referee #3:

The authors present a series of ultrahigh-resolution fluorescence studies in cell cultures using Airyscan and STED technology. They conclude that the electrochemical potential is heterogeneously distributed across the inner mitochondrial membrane. Specifically, cristae membranes carry a higher potential than the inner boundary membrane and different cristae within a single mitochondrion do not necessarily have identical potentials.

The manuscript is considerably more focused and contains more convincing data than a previous version that I was also asked to review. It starts out with a lucid, well-written account of the problem at hand (introduction) and continues with a large but well-organized set of data, which together in my opinion make a compelling case for cristae heterogeneity. In particular, the oligomycin/FCCP inhibitor experiments, the data on MICOS mutants, and the depolarization time series add depth to the manuscript.

The idea that individual cristae function relatively autonomously, if proven correct, may have fundamental implications for our understanding of mitochondrial physiology.

I have two points of criticism, which I believe need to be addressed and a few suggestions.

Major points

1. It is troubling to see that two potential-independent dyes (NAO and MTG) show the same heterogeneity between cristae and the IBM as the potential-dependent dyes (TMRE and Rho123) because it calls into question the central conclusion of the paper. Perhaps the explanation is that NAO stains cristae stronger than the IBM because cristae contain more cardiolipin. Likewise, MTG may stain cristae stronger than the IBM because cristae contain more proteins. However, these explanations remain conjectures at this point. The ideal control experiment would show similar fluorescence in cristae and IBM using a potential-independent dye.
2. As is the case for any truly novel concept, it is challenging to find the right language to convey the new idea. The opening sentence of the abstract, in which an analogy from electrical engineering is used, causes confusion right from the start. Although the "cable" idea is explained later in the paper, I think it is confusing in the specific context of the abstract because the principle direction of the flow of charges in mitochondria is perpendicular to the plane of the membrane whereas in a cable it is parallel to the long axis. If an analogy from electrical engineering were to be used, it

should be that of a capacitor. However, it might be better to stay away from such comparisons altogether. Furthermore, in the title the authors talk about "independent bioenergetic units". While the term is explained in the main body of the manuscript, the meaning of "independent bioenergetic unit" was not instantly obvious to me when I read the title and presumably will not be to others, because the term has not been defined. Consequently, it is not a good choice for the title.

Minor suggestions

3. Fig. 3A: Proper indices have to be included with the mathematical symbols of the equation. In its current form, $FL_{comp}/Fl_{comp}=1$, which makes $\Delta\Psi_m=0$.

4. The presentation of the data in Figs 6H is unnecessarily convoluted. I think it would be better to present the mean and the standard deviation of the $\Delta\Psi$'s of the individual cristae A, B, C, and D.

5. The sentence "The Nernst equation can be applied to the unbound portion of $\Delta\Psi_m$ probes (e.g., TMRE), where $\Delta\Psi_m$ differences across a membrane between two compartments can be calculated, since fluorescence intensities can be used to deduce fold differences in concentrations of the probe in the two compartments." (p. 10) does not make any sense to me.

6. The abbreviation LUT has to be defined.

7. P 9 bottom: "...depend on $\Delta\Psi_m$, we would..."

2nd Revision - authors' response

12th August 2019

Referee #1:

The manuscript has been thoroughly revised and my main general concerns have been effectively addressed experimentally. I would like to commend the authors for this excellent job. However, I still have a major concern (not requiring experiments) that I believe needs to be addressed.

We appreciate and are humbled by the assessment of the reviewer stating that we did an excellent job.

1) Beginning of the Introduction, the statement "The high concentration of hydrogen ions in the intermembrane space (IMS) constitutes the proton motive force (Δp), made up of electrical ($\Delta\Psi_m$) and chemical (ΔpH) potential energy (Mitchell, 1961)" is incorrect. Formation and maintenance of the mitochondrial proton gradient depends on both proton pumping and on the insulating properties of the inner membrane. The protonmotive force does not depend on the extramitochondrial (intermembrane space) concentration of protons. This is a misconception that in the early days was used as an argument against chemiosmosis. The comment was that proton pumping into the cytosol was like proton pumping into the ocean, implying that a proton gradient could not build up because of proton diffusion! The (still valid) point of Mitchell was that the proton gradient would form even if extramitochondrial protons were diluted into the ocean, because the extruded proton would still "leave behind" a more electronegative matrix (i.e., a membrane potential DIFFERENCE would still be formed because there is little or no permeability to anions and other charged species). The chemical (delta pH) and electrical (delta psi) components actually balance in opposite directions, and the membrane potential difference decreases when the delta pH increases (i.e., when a measurable acidification takes place like in the compartments described here).

We thank the reviewer for spotting this inaccuracy. We have removed all references and discussions on the Δp and only focused on $\Delta\Psi_m$, as this is the only bioenergetic parameter of the protonmotive force that we have measured in our study. The removed statement in the introduction was replaced by the following text: "Mitochondria utilize nutrients and molecular oxygen to generate a membrane potential ($\Delta\Psi_m$) across the inner mitochondrial membrane (IMM). The energy available for ATP synthesis is directly derived from $\Delta\Psi_m$; therefore, depolarization directly translates to decreased energy availability for ATP synthesis."

To further highlight that we grasp this point, we will paste the following passage from the textbook *Bioenergetics 4*, page 5, Chapter 1.1, which clearly summarizes why $\Delta\Psi_m$ is the most relevant parameter in mitochondria: "In only a few cases, such as the chloroplast, does Δp exist mainly as a pH difference across the energy-conserving membrane. In this example, the pH gradient, ΔpH , across the thylakoid membrane can exceed 3 units. Although the thylakoid space is therefore highly acidic, there are no enzymes in this compartment that might be compromised by the low pH. The more common situation is where $\Delta\Psi$ is the dominant component and the pH gradient is small,

perhaps only 0.5 pH units. This occurs, for example, in the mitochondrion, allowing enzymes in both the mitochondrial matrix and the cell cytoplasm to operate close to neutral pH.”

2) *A few lines below the authors state: "To date, the prevailing view of the Δp is that, after respiratory complexes I, III, and IV pump protons from the matrix across the IMM, the protons are free to move throughout the IMS (Amchenkova, 1988; Mannella et al., 2013; Skulachev, 2001). According to this model, random distribution of protons throughout the IMS would lead to an equal distribution of positive charges all along the outward-facing leaflet of the IMM. Consequently, the difference in proton concentration and, in turn, the difference in charge, between the IMS and matrix would be the same at any point along the IMM. This model provides the theoretical basis that the $\Delta\Psi_m$ constitutes an equipotential - i.e., each mitochondrion, at any given time, represents a uniform voltage." Again, this is not the point. The outer membrane is very permeable to protons, and therefore IMS protons equilibrate rapidly (in the absence of constraints; in isolated mitochondria pumped protons are readily measured in solution) and the delta pH is small (also due to buffering of matrix protons by phosphate and organic anions). In other words, the "equipotential" membrane would not rely on proton diffusion, but rather on the ultrastructure of mitochondria and on the homogeneity of the membrane properties (once the proton is extruded, the voltage "travels" along the membrane and is not "carried" by protons).*

We thank the reviewer for this comment, which is related to the inaccuracies raised in point number 1. We agree that the relevant parameter that reveals functional differences in mitochondrial function is the $\Delta\Psi_m$. Accordingly, in our study, we measured $\Delta\Psi_m$ and not the ΔpH , by using $\Delta\Psi_m$ -sensitive dyes. Thus, we removed this paragraph.

3) *Also, the discussion suffers from what appears to be a misconception. For example, page 22 "This study raises interesting questions as to why mitochondria organize the Δp in this way. One possibility is that the OMM is relatively porous compared to the IMM. If protons pumped across IMM by the respiratory complexes were permitted to diffuse throughout the IMS, it is likely that they would readily escape into the cytosol. Confining protons in the cristae lumen by increasing resistance at the CJ would, on the other hand, concentrate protons near the array of FoF₁ ATP Synthase dimers, increasing the probability of proton translocation back into the matrix and, in turn, promoting ATP production." ATP production is driven as easily by an electrical potential difference, which drags back the protons irrespective of the delta pH, so this is not a valid argument. What drives ATP synthesis is the proton electrochemical GRADIENT, not only the delta pH.*

We agree with the reviewer that the dominant force drawing protons back into the matrix, promoting ATP production, is the $\Delta\Psi_m$ rather than the ΔpH . Since we did not measure the ΔpH in this study, we removed this paragraph and substituted it with: “This study raises interesting questions as to why mitochondria organize the $\Delta\Psi_m$ in this way. One advantage, for example, could be related to the fact that $\Delta\Psi_m$ constitutes the main energy available to drive protons through F₁F_o ATP Synthase to produce ATP. As such, the localization of F₁F_o ATP Synthase at the cristae rims appears to be advantageous in terms of proximity to the batteries.”

4) *Also at page 22, "Another possibility that the IMM has evolved different electrical domains could be to regulate import and export of proteins, metabolites, ions, etc. An equipotential could interfere with the movement of various essential molecules in and out of the organelle: if there were an equally high proton concentration at any point throughout the IMS, then the relatively high concentration of positive charges could act as an electrical barrier against other positively charged molecules or ions needing to cross the membrane. A hetero-potential shows decreased proton concentration near the IBM, which could decrease resistance to transport of various materials in and out of the organelle." These speculations are excessive, and I don't think that they are needed. It is fine to say that physical constraints between cristae may lead to individual "units" (as measured here), but the acidification of these compartments (not measured here) would be the consequence, not the cause of the heterogeneity. And it should be noted that the higher the delta pH (i.e. the local acidification) the lower the delta psi and vice versa. I think that the "barriers" at the crista junctions should rather be viewed as electrical barriers, similar to myelinated regions of the neuron, i.e. structures that prevent lateral propagation of the electrical signal to the next structure. This may well be (and probably is) a dynamic process. Certainly heterogeneity opens the door to subtle changes in microdomains that may then fulfill different local requirements. What is the meaning is hard to guess at present, but I feel that speculations should be kept to a minimum.*

We thank the reviewer for indicating that we should minimize our speculations, which are now confined to:

“Hyperpolarized and depolarized IMM potentials are associated with different states of respiration. While both uncoupling and an increased rate of ATP synthesis dissipate $\Delta\Psi_m$, a decrease in ATP synthesis may result in hyperpolarization and increased ROS production. The hetero-potential model of the mitochondrion allows for different cristae to serve different functions. In this model, some cristae could be more dedicated to ATP synthesis, whereas neighboring cristae could play a role in ROS signaling. The hetero-potential model further allows for the consideration that different cristae may engage in primarily complex II vs. complex I respiration, which are associated with different membrane potentials and could be driven by different fuels.”

Referee #3:

The authors present a series of ultrahigh-resolution fluorescence studies in cell cultures using Airyscan and STED technology. They conclude that the electrochemical potential is heterogeneously distributed across the inner mitochondrial membrane. Specifically, cristae membranes carry a higher potential than the inner boundary membrane and different cristae within a single mitochondrion do not necessarily have identical potentials.

The manuscript is considerably more focused and contains more convincing data than a previous version that I was also asked to review. It starts out with a lucid, well-written account of the problem at hand (introduction) and continues with a large but well-organized set of data, which together in my opinion make a compelling case for cristae heterogeneity. In particular, the oligomycin/FCCP inhibitor experiments, the data on MICOS mutants, and the depolarization time series add depth to the manuscript.

The idea that individual cristae function relatively autonomously, if proven correct, may have fundamental implications for our understanding of mitochondrial physiology.

We thank the reviewer for stating that our study has been improved, containing more convincing data and that we provide a compelling case for functional heterogeneity in cristae.

I have two points of criticism, which I believe need to be addressed and a few suggestions.

Major points

1. It is troubling to see that two potential-independent dyes (NAO and MTG) show the same heterogeneity between cristae and the IBM as the potential-dependent dyes (TMRE and Rho123) because it calls into question the central conclusion of the paper. Perhaps the explanation is that NAO stains cristae stronger than the IBM because cristae contain more cardiolipin. Likewise, MTG may stain cristae stronger than the IBM because cristae contain more proteins. However, these explanations remain conjectures at this point. The ideal control experiment would show similar fluorescence in cristae and IBM using a potential-independent dye.

We thank the reviewer for this comment. While we highlighted that MTG requires membrane potential to localize to the mitochondria, meaning that more MTG will accumulate in areas with higher $\Delta\Psi_m$ potential, it is also true that differences in protein and lipid composition along the IMM could also explain the differences in MTG/NAO binding and thus staining. To acknowledge this, we have included the following paragraph in the discussion: *“A limitation of this methodology is the lack of dyes that are completely independent of $\Delta\Psi_m$ or cardiolipin content. As a result, the NAO and MTG staining of IBM appeared weaker than that of cristae, precluding ratiometric imaging. This limitation only pertains to the differences in $\Delta\Psi_m$ of cristae vs. IBM but would not affect conclusions related to the $\Delta\Psi_m$ differences between different cristae within the same mitochondrion.”*

On the other hand, our data quantifying just the signal from the diffusible fraction of TMRE (non-binding and thus sensitive to $\Delta\Psi_m$) which also responds differently in the cristae and IBM to oligomycin and FCCP, strongly supports that increased diffusible TMRE FI in the cristae is caused by increased $\Delta\Psi_m$, when compared to the IBM.

2. As is the case for any truly novel concept, it is challenging to find the right language to convey the new idea. The opening sentence of the abstract, in which an analogy from electrical engineering is used, causes confusion right from the start. Although the "cable" idea is explained later in the paper, I think it is confusing in the specific context of the abstract because the principle direction of the flow of charges in mitochondria is perpendicular to the

plane of the membrane whereas in a cable it is parallel to the long axis. If an analogy from electrical engineering were to be used, it should be that of a capacitor. However, it might be better to stay away from such comparisons altogether. Furthermore, in the title the authors talk about "independent bioenergetic units". While the term is explained in the main body of the manuscript, the meaning of "independent bioenergetic unit" was not instantly obvious to me when I read the title and presumably will not be to others, because the term has not been defined. Consequently, it is not a good choice for the title.

We thank the reviewer for such an insightful comment. To address this point, we changed the title to: *Individual cristae within the single mitochondrion have different membrane potentials, functioning as independent bioenergetic units*

The cable idea has been removed from the abstract and now we only introduce the concept of bioenergetic independency after explaining that different cristae have differences in $\Delta\Psi_m$, responding differently to oligomycin and FCCP, which introduces better the idea of independent bioenergetic units:

"The mitochondrial membrane potential ($\Delta\Psi_m$) is the main driver of OXPHOS. The inner mitochondrial membrane (IMM), consisting of cristae and inner boundary membranes (IBM), is considered to carry a uniform $\Delta\Psi_m$. However, sequestration of OXPHOS components in cristae membranes necessitates a re-examination of the equipotential representation of the IMM. We developed an approach to monitor $\Delta\Psi_m$ at the resolution of individual cristae. We found that the IMM was divided into segments with distinct $\Delta\Psi_m$, corresponding to cristae and IBM. $\Delta\Psi_m$ was higher at cristae compared to IBM. Treatment with oligomycin increased, whereas FCCP decreased, $\Delta\Psi_m$ heterogeneity along the IMM. Impairment of cristae structure through deletion of MICOS-complex components or Opa1 diminished this intramitochondrial heterogeneity of $\Delta\Psi_m$. Lastly, we determined that different cristae within the individual mitochondrion can have disparate membrane potentials and that interventions causing acute depolarization may affect some cristae while sparing others. Altogether, our data support a new model in which cristae within the same mitochondrion behave as independent bioenergetic units, preventing the failure of specific cristae from spreading dysfunction to the rest."

Finally, we followed the suggestion of the reviewer and, in the discussion, we included an analogy to a capacitor to describe the equipotential model:

"In the case of the equipotential model, where the inner membrane of the entire mitochondrion represents a single capacitor, a breach in membrane integrity in one crista would cause a collapse in voltage in all cristae and compromise the function of the whole organelle."

Minor suggestions

3. Fig. 3A: Proper indices have to be included with the mathematical symbols of the equation. In its current form, $FL_{comp}/FI_{comp}=1$, which makes $\Delta\Psi_m=0$.

We thank the reviewer for spotting this. We included now in the figure $FI_{comp}/FL_{comp}=1$, $\Delta\Psi_m=0$, which now better illustrates that differences in TMRE FI reveal differences in $\Delta\Psi_m$.

4. The presentation of the data in Figs 6H is unnecessarily convoluted. I think it would be better to present the mean and the standard deviation of the $\Delta\Psi$'s of the individual cristae A, B, C, and D.

We thank the reviewer for this suggestion clarifying our findings. This suggestion is now new Figure panel 10B.

5. The sentence "The Nernst equation can be applied to the unbound portion of $\Delta\Psi_m$ probes (e.g., TMRE), where $\Delta\Psi_m$ differences across a membrane between two compartments can be calculated, since fluorescence intensities can be used to deduce fold differences in concentrations of the probe in the two compartments." (p. 10) does not make any sense to me.

We apologize for the lack of clarity of this sentence. Now it reads: *"The Nernst equation can be used to quantify $\Delta\Psi_m$ by acquiring the FI of $\Delta\Psi_m$ -sensitive probes (e.g., TMRE). The FIs of the probes at different subcellular compartments can be used to extrapolate the differences in concentrations of the probe, which are needed to calculate the difference in $\Delta\Psi_m$ between compartments (Ehrenberg et al., 1988; Farkas et al., 1989; Loew et al., 1993; Twig et al., 2008; Wikstrom et al., 2007)."*

6. The abbreviation LUT has to be defined.

This has been defined: LookUp Table (LUT)

7. P 9 bottom: "...depend on $\Delta\Psi_m$, we would..."

We thank the reviewer for spotting this typo. We corrected it, as suggested above: *“Moreover, if the differences in TMRE fluorescence intensity (FI) between cristae and IBM depend on $\Delta\Psi_m$, we would expect that the differences between the brightest and dimmest pixels would markedly decrease during depolarization”*.

Accepted

5th September 2019

I am pleased to inform you that your manuscript has been accepted for publication in the EMBO Journal.

Corresponding Author Name: Marc Liesa and Orian S. Shirihai

Journal Submitted to: EMBO J

Manuscript Number: EMBOJ-2018-101056R1